# Rapidly Mixing Gibbs Sampling for a Class of Factor Graphs Using Hierarchy Width

**Christopher De Sa, Ce Zhang, Kunle Olukotun, and Christopher Ré**
cdesa@stanford.edu, czhang@cs.wisc.edu,
kunle@stanford.edu, chrismre@stanford.edu
Departments of Electrical Engineering and Computer Science
Stanford University, Stanford, CA 94309

## Abstract

Gibbs sampling on factor graphs is a widely used inference technique, which often produces good empirical results. Theoretical guarantees for its performance are weak: even for tree structured graphs, the mixing time of Gibbs may be exponential in the number of variables. To help understand the behavior of Gibbs sampling, we introduce a new (hyper)graph property, called *hierarchy width*. We show that under suitable conditions on the weights, bounded hierarchy width ensures polynomial mixing time. Our study of hierarchy width is in part motivated by a class of factor graph templates, *hierarchical templates*, which have bounded hierarchy width—regardless of the data used to instantiate them. We demonstrate a rich application from natural language processing in which Gibbs sampling provably mixes rapidly and achieves accuracy that exceeds human volunteers.

## 1 Introduction

We study inference on factor graphs using Gibbs sampling, the de facto Markov Chain Monte Carlo (MCMC) method [8, p. 505]. Specifically, our goal is to compute the marginal distribution of some *query* variables using Gibbs sampling, given *evidence* about some other variables and a set of factor weights. We focus on the case where all variables are discrete. In this situation, a Gibbs sampler randomly updates a single variable at each iteration by sampling from its conditional distribution given the values of all the other variables in the model. Many systems—such as Factorie [14], OpenBugs [12], PGibbs [5], DimmWitted [28], and others [15, 22, 25]—use Gibbs sampling for inference because it is fast to run, simple to implement, and often produces high quality empirical results. However, theoretical guarantees about Gibbs are lacking. The aim of the technical result of this paper is to provide new cases in which one can guarantee that Gibbs gives accurate results.

For an MCMC sampler like Gibbs sampling, the standard measure of efficiency is the *mixing time* of the underlying Markov chain. We say that a Gibbs sampler *mixes rapidly* over a class of models if its mixing time is at most polynomial in the number of variables in the model. Gibbs sampling is known to mix rapidly for some models. For example, Gibbs sampling on the Ising model on a graph with bounded degree is known to mix in quasilinear time for high temperatures [10, p. 201]. Recent work has outlined conditions under which Gibbs sampling of Markov Random Fields mixes rapidly [11]. Continuous-valued Gibbs sampling over models with exponential-family distributions is also known to mix rapidly [2, 3]. Each of these celebrated results still leaves a gap: there are many classes of factor graphs on which Gibbs sampling seems to work very well—including as part of systems that have won quality competitions [24]—for which there are no theoretical guarantees of rapid mixing.

Many graph algorithms that take exponential time in general can be shown to run in polynomial time as long as some graph property is bounded. For inference on factor graphs, the most commonly

used property is hypertree width, which bounds the complexity of *dynamic programming* algorithms on the graph. Many problems, including variable elimination for exact inference, can be solved in polynomial time on graphs with bounded hypertree width [8, p. 1000]. In some sense, bounded hypertree width is a necessary and sufficient condition for tractability of inference in graphical models [1, 9]. Unfortunately, it is not hard to construct examples of factor graphs with bounded weights and hypertree width 1 for which Gibbs sampling takes exponential time to mix. Therefore, bounding hypertree width is insufficient to ensure rapid mixing of Gibbs sampling. To analyze the behavior of Gibbs sampling, we define a new graph property, called the *hierarchy width*. This is a stronger condition than hypertree width; the hierarchy width of a graph will always be larger than its hypertree width. We show that for graphs with bounded hierarchy width and bounded weights, Gibbs sampling mixes rapidly.

Our interest in hierarchy width is motivated by so-called factor graph templates, which are common in practice [8, p. 213]. Several types of models, such as Markov Logic Networks (MLN) and Relational Markov Networks (RMN) can be represented as factor graph templates. Many state-of-the-art systems use Gibbs sampling on factor graph templates and achieve better results than competitors using other algorithms [14, 27]. We exhibit a class of factor graph templates, called *hierarchical templates*, which, when instantiated, have a hierarchy width that is bounded independently of the dataset used; Gibbs sampling on models instantiated from these factor graph templates will mix in polynomial time. This is a kind of sampling analog to tractable Markov logic [4] or so-called "safe plans" in probabilistic databases [23]. We exhibit a real-world templated program that outperforms human annotators at a complex text extraction task—and provably mixes in polynomial time.

In summary, this work makes the following contributions:

- We introduce a new notion of width, *hierarchy width*, and show that Gibbs sampling mixes in polynomial time for all factor graphs with bounded hierarchy width and factor weight.
- We describe a new class of factor graph templates, *hierarchical factor graph templates*, such that Gibbs sampling on instantiations of these templates mixes in polynomial time.
- We validate our results experimentally and exhibit factor graph templates that achieve high quality on tasks but for which our new theory is able to provide mixing time guarantees.

## 1.1   Related Work

Gibbs sampling is just one of several algorithms proposed for use in factor graph inference. The variable elimination algorithm [8] is an exact inference method that runs in polynomial time for graphs of bounded hypertree width. Belief propagation is another widely-used inference algorithm that produces an exact result for trees and, although it does not converge in all cases, converges to a good approximation under known conditions [7]. *Lifted inference* [18] is one way to take advantage of the structural symmetry of factor graphs that are instantiated from a template; there are lifted versions of many common algorithms, such as variable elimination [16], belief propagation [21], and Gibbs sampling [26]. It is also possible to leverage a template for fast computation: Venugopal et al. [27] achieve orders of magnitude of speedup of Gibbs sampling on MLNs. Compared with Gibbs sampling, these inference algorithms typically have better theoretical results; despite this, Gibbs sampling is a ubiquitous algorithm that performs practically well—far outstripping its guarantees.

Our approach of characterizing runtime in terms of a graph property is typical for the analysis of graph algorithms. Many algorithms are known to run in polynomial time on graphs of bounded treewidth [19], despite being otherwise NP-hard. Sometimes, using a stronger or weaker property than treewidth will produce a better result; for example, the submodular width used for constraint satisfaction problems [13].

## 2   Main Result

In this section, we describe our main contribution. We analyze some simple example graphs, and use them to show that bounded hypertree width is not sufficient to guarantee rapid mixing of Gibbs sampling. Drawing intuition from this, we define the hierarchy width graph property, and prove that Gibbs sampling mixes in polynomial time for graphs with bounded hierarchy width.

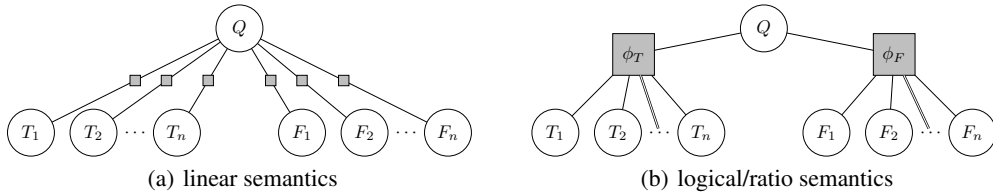

<div align="center">(a) linear semantics        (b) logical/ratio semantics</div>

Figure 1: Factor graph diagrams for the voting model; single-variable prior factors are omitted.

First, we state some basic definitions. A factor graph $G$ is a graphical model that consists of a set of variables $V$ and factors $\Phi$, and determines a distribution over those variables. If $I$ is a *world* for $G$ (an assignment of a value to each variable in $V$), then $\epsilon$, the *energy* of the world, is defined as

$$\epsilon(I) = \sum_{\phi \in \Phi} \phi(I). \tag{1}$$

The probability of world $I$ is $\pi(I) = \frac{1}{Z} \exp(\epsilon(I))$, where $Z$ is the normalization constant necessary for this to be a distribution. Typically, each $\phi$ depends only on a subset of the variables; we can draw $G$ as a bipartite graph where a variable $v \in V$ is connected to a factor $\phi \in \Phi$ if $\phi$ depends on $v$.

**Definition 1** (Mixing Time). The *mixing time* of a Markov chain is the first time $t$ at which the estimated distribution $\mu_t$ is within statistical distance $\frac{1}{4}$ of the true distribution [10, p. 55]. That is,

$$t_{\text{mix}} = \min \left\{ t : \max_{A \subset \Omega} |\mu_t(A) - \pi(A)| \leq \tfrac{1}{4} \right\}.$$

## 2.1 Voting Example

We start by considering a simple example model [20], called the *voting model*, that models the sign of a particular "query" variable $Q \in \{-1, 1\}$ in the presence of other "voter" variables $T_i \in \{0, 1\}$ and $F_i \in \{0, 1\}$, for $i \in \{1, \ldots, n\}$, that suggest that $Q$ is positive and negative (true and false), respectively. We consider three versions of this model. The first, the *voting model with linear semantics*, has energy function

$$\epsilon(Q, T, F) = wQ \sum_{i=1}^{n} T_i - wQ \sum_{i=1}^{n} F_i + \sum_{i=1}^{n} w_{T_i} T_i + \sum_{i=1}^{n} w_{F_i} F_i,$$

where $w_{T_i}$, $w_{F_i}$, and $w > 0$ are constant weights. This model has a factor connecting each voter variable to the query, which represents the value of that vote, and an additional factor that gives a prior for each voter. It corresponds to the factor graph in Figure 1(a). The second version, the *voting model with logical semantics*, has energy function

$$\epsilon(Q, T, F) = wQ \max_i T_i - wQ \max_i F_i + \sum_{i=1}^{n} w_{T_i} T_i + \sum_{i=1}^{n} w_{F_i} F_i.$$

Here, in addition to the prior factors, there are only two other factors, one of which (which we call $\phi_T$) connects all the true-voters to the query, and the other of which ($\phi_F$) connects all the false-voters to the query. The third version, the *voting model with ratio semantics*, is an intermediate between these two models, and has energy function

$$\epsilon(Q, T, F) = wQ \log \left(1 + \sum_{i=1}^{n} T_i\right) - wQ \log \left(1 + \sum_{i=1}^{n} F_i\right) + \sum_{i=1}^{n} w_{T_i} T_i + \sum_{i=1}^{n} w_{F_i} F_i.$$

With either logical or ratio semantics, this model can be drawn as the factor graph in Figure 1(b).

These three cases model different distributions and therefore different ways of representing the power of a vote; the choice of names is motivated by considering the marginal odds of $Q$ given the other variables. For linear semantics, the odds of $Q$ depend *linearly* on the difference between the number of nonzero positive-voters $T_i$ and nonzero negative-voters $F_i$. For ratio semantics, the odds of $Q$ depend roughly on their *ratio*. For logical semantics, only the presence of nonzero voters matters, not the number of voters.

We instantiated this model with random weights $w_{T_i}$ and $w_{F_i}$, ran Gibbs sampling on it, and computed the variance of the estimated marginal probability of $Q$ for the different models (Figure 2). The results show that the models with logical and ratio semantics produce much lower-variance estimates than the model with linear semantics. This experiment motivates us to try to prove a bound on the mixing time of Gibbs sampling on this model.

**Theorem 1.** *Fix any constant $\omega > 0$, and run Gibbs sampling on the voting model with bounded factor weights $\{w_{T_i}, w_{F_i}, w\} \subset [-\omega, \omega]$. For the voting model with linear semantics, the largest*

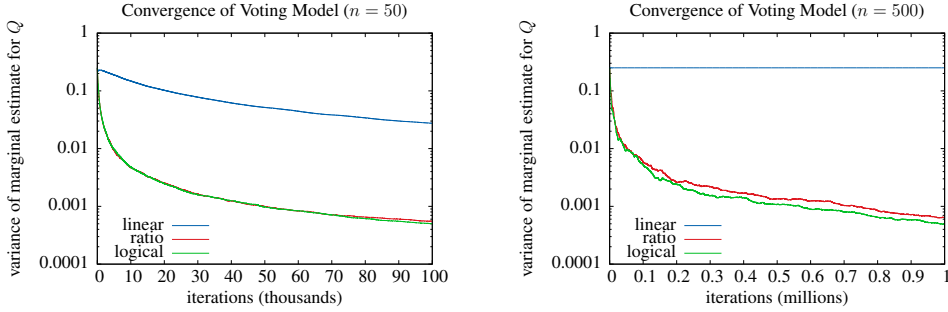

Figure 2: Convergence for the voting model with $w = 0.5$, and random prior weights in $(-1, 0)$.

*possible mixing time $t_{\mathrm{mix}}$ of any such model is $t_{\mathrm{mix}} = 2^{\Theta(n)}$. For the voting model with either logical or ratio semantics, the largest possible mixing time is $t_{\mathrm{mix}} = \Theta(n \log n)$.*

This result validates our observation that linear semantics mix poorly compared to logical and ratio semantics. Intuitively, the reason why linear semantics performs worse is that the Gibbs sampler will switch the state of $Q$ only very infrequently—in fact exponentially so. This is because the energy roughly depends linearly on the number of voters $n$, and therefore the probability of switching $Q$ depends exponentially on $n$. This does not happen in either the logical or ratio models.

## 2.2 Hypertree Width

In this section, we describe the commonly-used graph property of hypertree width, and show using the voting example that bounding it is insufficient to ensure rapid Gibbs sampling. Hypertree width is typically used to bound the complexity of dynamic programming algorithms on a graph; in particular, variable elimination for exact inference runs in polynomial time on factor graphs with bounded hypertree width [8, p. 1000]. The hypertree width of a hypergraph, which we denote $\mathrm{tw}(G)$, is a generalization of the notion of acyclicity; since the definition of hypertree width is technical, we instead state the definition of an acyclic hypergraph, which is sufficient for our analysis. In order to apply these notions to factor graphs, we can represent a factor graph as a hypergraph that has one vertex for each node of the factor graph, and one hyperedge for each factor, where that hyperedge contains all variables the factor depends on.

**Definition 2** (Acyclic Factor Graph [6]). A *join tree*, also called a junction tree, of a factor graph $G$ is a tree $T$ such that the nodes of $T$ are the factors of $G$ and, if two factors $\phi$ and $\rho$ both depend on the same variable $x$ in $G$, then every factor on the unique path between $\phi$ and $\rho$ in $T$ also depends on $x$. A factor graph is *acyclic* if it has a join tree. All acyclic graphs have hypertree width $\mathrm{tw}(G) = 1$.

Note that all trees are acyclic; in particular the voting model (with any semantics) has hypertree width 1. Since the voting model with linear semantics and bounded weights mixes in exponential time (Theorem 1), this means that bounding the hypertree width and the factor weights is insufficient to ensure rapid mixing of Gibbs sampling.

## 2.3 Hierarchy Width

Since the hypertree width is insufficient, we define a new graph property, the *hierarchy width*, which, when bounded, ensures rapid mixing of Gibbs sampling. This result is our main contribution.

**Definition 3** (Hierarchy Width). The hierarchy width $\mathrm{hw}(G)$ of a factor graph $G$ is defined recursively such that, for any *connected* factor graph $G = \langle V, \Phi \rangle$,

$$\mathrm{hw}(G) = 1 + \min_{\phi^* \in \Phi} \mathrm{hw}(\langle V, \Phi - \{\phi^*\}\rangle), \tag{2}$$

and for any *disconnected* factor graph $G$ with connected components $G_1, G_2, \ldots,$

$$\mathrm{hw}(G) = \max_i \mathrm{hw}(G_i). \tag{3}$$

As a *base case*, all factor graphs $G$ with no factors have

$$\mathsf{hw}(\langle V, \emptyset \rangle) = 0. \tag{4}$$

To develop some intuition about how to use the definition of hierarchy width, we derive the hierarchy width of the path graph drawn in Figure 3.

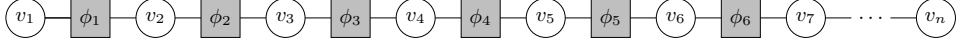

Figure 3: Factor graph diagram for an $n$-variable path graph.

**Lemma 1.** *The path graph model has hierarchy width $\mathsf{hw}(G) = \lceil \log_2 n \rceil$.*

*Proof.* Let $G_n$ denote the path graph with $n$ variables. For $n = 1$, the lemma follows from (4). For $n > 1$, $G_n$ is connected, so we must compute its hierarchy width by applying (2). It turns out that the factor that minimizes this expression is the factor in the middle, and so applying (2) followed by (3) shows that $\mathsf{hw}(G_n) = 1 + \mathsf{hw}(\mathsf{G}_{\lceil \frac{n}{2} \rceil})$. Applying this inductively proves the lemma. $\square$

Similarly, we are able to compute the hierarchy width of the voting model factor graphs.

**Lemma 2.** *The voting model with logical or ratio semantics has hierarchy width $\mathsf{hw}(G) = 3$.*

**Lemma 3.** *The voting model with linear semantics has hierarchy width $\mathsf{hw}(G) = 2n + 1$.*

These results are promising, since they separate our polynomially-mixing examples from our exponentially-mixing examples. However, the hierarchy width of a factor graph says nothing about the factors themselves and the functions they compute. This means that it, alone, tells us nothing about the model; for example, any distribution can be represented by a trivial factor graph with a single factor that contains all the variables. Therefore, in order to use hierarchy width to produce a result about the mixing time of Gibbs sampling, we constrain the maximum weight of the factors.

**Definition 4** (Maximum Factor Weight)**.** A factor graph has maximum factor weight $M$, where

$$M = \max_{\phi \in \Phi} \left( \max_I \phi(I) - \min_I \phi(I) \right).$$

For example, the maximum factor weight of the voting example with linear semantics is $M = 2w$; with logical semantics, it is $M = 2w$; and with ratio semantics, it is $M = 2w \log(n + 1)$. We now show that graphs with bounded hierarchy width and maximum factor weight mix rapidly.

**Theorem 2** (Polynomial Mixing Time)**.** *If $G$ is a factor graph with $n$ variables, at most $s$ states per variable, $e$ factors, maximum factor weight $M$, and hierarchy width $h$, then*

$$t_{\mathrm{mix}} \le (\log(4) + n \log(s) + eM) \, n \exp(3hM).$$

*In particular, if $e$ is polynomial in $n$, the number of values for each variable is bounded, and $hM = O(\log n)$, then $t_{\mathrm{mix}}(\epsilon) = O(n^{O(1)})$.*

To show why bounding the hierarchy width is necessary for this result, we outline the proof of Theorem 2. Our technique involves bounding the absolute spectral gap $\gamma(G)$ of the transition matrix of Gibbs sampling on graph $G$; there are standard results that use the absolute spectral gap to bound the mixing time of a process [10, p. 155]. Our proof proceeds via induction using the definition of hierarchy width and the following three lemmas.

**Lemma 4** (Connected Case)**.** *Let $G$ and $\bar{G}$ be two factor graphs with maximum factor weight $M$, which differ only inasmuch as $G$ contains a single additional factor $\phi^*$. Then,*

$$\gamma(G) \ge \gamma(\bar{G}) \exp(-3M).$$

**Lemma 5** (Disconnected Case)**.** *Let $G$ be a disconnected factor graph with $n$ variables and $m$ connected components $G_1, G_2, \ldots, G_m$ with $n_1, n_2, \ldots n_m$ variables, respectively. Then,*

$$\gamma(G) \ge \min_{i \le m} \frac{n_i}{n} \gamma(G_i).$$

**Lemma 6** (Base Case). *Let $G$ be a factor graph with one variable and no factors. The absolute spectral gap of Gibbs sampling running on $G$ will be $\gamma(G) = 1$.*

Using these Lemmas inductively, it is not hard to show that, under the conditions of Theorem 2,

$$\gamma(G) \geq \frac{1}{n} \exp\left(-3hM\right);$$

converting this to a bound on the mixing time produces the result of Theorem 2.

To gain more intuition about the hierarchy width, we compare its properties to those of the hypertree width. First, we note that, when the hierarchy width is bounded, the hypertree width is also bounded.

**Statement 1.** *For any factor graph $G$, $\mathsf{tw}(G) \leq \mathsf{hw}(G)$.*

One of the useful properties of the hypertree width is that, for any fixed $k$, computing whether a graph $G$ has hypertree width $\mathsf{tw}(G) \leq k$ can be done in polynomial time in the size of $G$. We show the same is true for the hierarchy width.

**Statement 2.** *For any fixed $k$, computing whether $\mathsf{hw}(G) \leq k$ can be done in time polynomial in the number of factors of $G$.*

Finally, we note that we can also bound the hierarchy width using the degree of the factor graph. Notice that a graph with unbounded node degree contains the voting program with linear semantics as a subgraph. This statement shows that bounding the hierarchy width disallows such graphs.

**Statement 3.** *Let $d$ be the maximum degree of a variable in factor graph $G$. Then, $\mathsf{hw}(G) \geq d$.*

## 3   Factor Graph Templates

Our study of hierarchy width is in part motivated by the desire to analyze the behavior of Gibbs sampling on factor graph templates, which are common in practice and used by many state-of-the-art systems. A factor graph template is an abstract model that can be *instantiated* on a dataset to produce a factor graph. The dataset consists of *objects*, each of which represents a thing we want to reason about, which are divided into *classes*. For example, the object Bart could have class Person and the object Twilight could have class Movie. (There are many ways to define templates; here, we follow the formulation in Koller and Friedman [8, p. 213].)

A factor graph template consists of a set of template variables and template factors. A template variable represents a property of a tuple of zero or more objects of particular classes. For example, we could have an IsPopular$(x)$ template, which takes a single argument of class Movie. In the instantiated graph, this would take the form of multiple variables like IsPopular(Twilight) or IsPopular(Avengers). Template factors are replicated similarly to produce multiple factors in the instantiated graph. For example, we can have a template factor

$$\phi\left(\mathsf{TweetedAbout}(x, y), \mathsf{IsPopular}(x)\right)$$

for some factor function $\phi$. This would be instantiated to factors like

$$\phi\left(\mathsf{TweetedAbout}(\mathsf{Avengers}, \mathsf{Bart}), \mathsf{IsPopular}(\mathsf{Avengers})\right).$$

We call the $x$ and $y$ in a template factor *object symbols*. For an instantiated factor graph with template factors $\Phi$, if we let $A_\phi$ denote the set of possible assignments to the object symbols in a template factor $\phi$, and let $\phi(a, I)$ denote the value of its factor function in world $I$ under the object symbol assignment $a$, then the standard way to define the energy function is with

$$\epsilon(I) = \sum_{\phi \in \Phi} \sum_{a \in A_\phi} w_\phi \phi(a, I), \tag{5}$$

where $w_\phi$ is the weight of template factor $\phi$. This energy function results from the creation of a single factor $\phi_a(I) = \phi(a, I)$ for each object symbol assignment $a$ of $\phi$. Unfortunately, this standard energy definition is not suitable for all applications. To deal with this, Shin et al. [20] introduce the notion of a *semantic function $g$*, which counts the of energy of instances of the factor template in a non-standard way. In order to do this, they first divide the object symbols of each template factor into two groups, the *head symbols* and the *body symbols*. When writing out factor templates, we distinguish head symbols by writing them with a hat (like $\hat{x}$). If we let $H_\phi$ denote the set of possible assignments to the head symbols, let $B_\phi$ denote the set of possible assignments

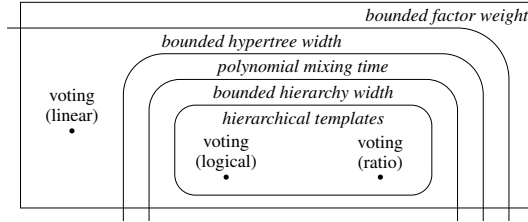

Figure 4: Subset relationships among classes of factor graphs, and locations of examples.

to the body symbols, and let $\phi(h, b, I)$ denote the value of its factor function in world $I$ under the assignment $(h, b)$, then the energy of a world is defined as

$$\epsilon(I) = \sum_{\phi \in \Phi} \sum_{h \in H_\phi} w_\phi(h)\, g\left(\sum_{b \in B_\phi} \phi(h, b, I)\right). \tag{6}$$

This results in the creation of a single factor $\phi_h(I) = g\left(\sum_b \phi(h, b, I)\right)$ for each assignment of the template's head symbols. We focus on three semantic functions in particular [20]. For the first, *linear semantics*, $g(x) = x$. This is identical to the standard semantics in (5). For the second, *logical semantics*, $g(x) = \text{sgn}(x)$. For the third, *ratio semantics*, $g(x) = \text{sgn}(x) \log(1 + |x|)$. These semantics are analogous to the different semantics used in our voting example. Shin et al. [20] exhibit several classification problems where using logical or ratio semantics gives better F1 scores.

### 3.1 Hierarchical Factor Graphs

In this section, we outline a class of templates, hierarchical templates, that have bounded hierarchy width. We focus on models that have hierarchical structure in their template factors; for example,

$$\phi(A(\hat{x}, \hat{y}, z), B(\hat{x}, \hat{y}), Q(\hat{x}, \hat{y})) \tag{7}$$

should have hierarchical structure, while

$$\phi(A(z), B(\hat{x}), Q(\hat{x}, y)) \tag{8}$$

should not. Armed with this intuition, we give the following definitions.

**Definition 5** (Hierarchy Depth)**.** A template factor $\phi$ has *hierarchy depth* $d$ if the first $d$ object symbols that appear in each of its terms are the same. We call these symbols *hierarchical symbols*. For example, (7) has hierarchy depth 2, and $\hat{x}$ and $\hat{y}$ are hierarchical symbols; also, (8) has hierarchy depth 0, and no hierarchical symbols.

**Definition 6** (Hierarchical)**.** We say that a template factor is *hierarchical* if all of its head symbols are hierarchical symbols. For example, (7) is hierarchical, while (8) is not. We say that a factor graph template is *hierarchical* if all its template factors are hierarchical.

We can explicitly bound the hierarchy width of instances of hierarchical factor graphs.

**Lemma 7.** *If $G$ is an instance of a hierarchical template with $E$ template factors, then $\text{hw}(G) \leq E$.*

We would now like to use Theorem 2 to prove a bound on the mixing time; this requires us to bound the maximum factor weight of the graph. Unfortunately, for linear semantics, the maximum factor weight of a graph is potentially $O(n)$, so applying Theorem 2 won't get us useful results. Fortunately, for logical or ratio semantics, hierarchical factor graphs do mix in polynomial time.

**Statement 4.** *For any fixed hierarchical factor graph template $\mathcal{G}$, if $G$ is an instance of $\mathcal{G}$ with bounded weights using either logical or ratio semantics, then the mixing time of Gibbs sampling on $G$ is polynomial in the number of objects $n$ in its dataset. That is, $t_{\text{mix}} = O\left(n^{O(1)}\right)$.*

So, if we want to construct models with Gibbs samplers that mix rapidly, one way to do it is with hierarchical factor graph templates using logical or ratio semantics.

## 4 Experiments

**Synthetic Data** We constructed a synthetic dataset by using an ensemble of Ising model graphs each with 360 nodes, 359 edges, and treewidth 1, but with different hierarchy widths. These graphs

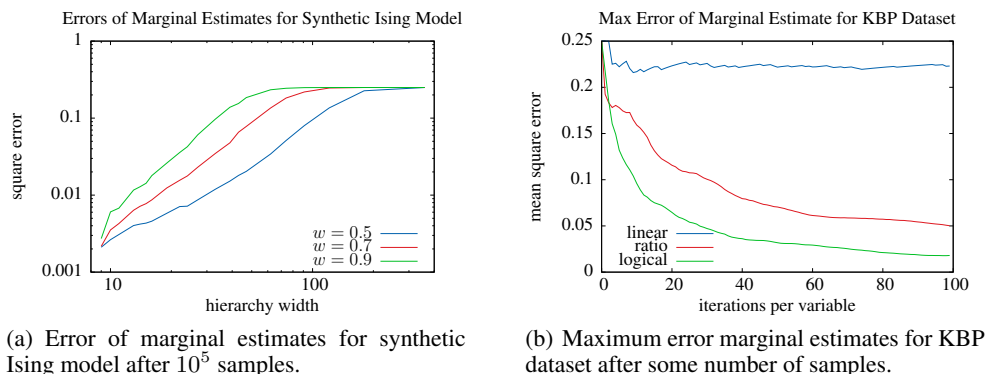

(a) Error of marginal estimates for synthetic Ising model after $10^5$ samples.

(b) Maximum error marginal estimates for KBP dataset after some number of samples.

Figure 5: Experiments illustrate how convergence is affected by hierarchy width and semantics.

ranged from the star graph (like in Figure 1(a)) to the path graph; and each had different hierarchy width. For each graph, we were able to calculate the exact true marginal of each variable because of the small tree-width. We then ran Gibbs sampling on each graph, and calculated the error of the marginal estimate of a single arbitrarily-chosen query variable. Figure 5(a) shows the result with different weights and hierarchy width. It shows that, even for tree graphs with the same number of nodes and edges, the mixing time can still vary depending on the hierarchy width of the model.

**Real-World Applications**   We observed that the hierarchical templates that we focus on in this work appear frequently in real applications. For example, all five knowledge base population (KBP) systems illustrated by Shin et al. [20] contain subgraphs that are grounded by hierarchical templates. Moreover, sometimes a factor graph is solely grounded by hierarchical templates, and thus provably mixes rapidly by our theorem while achieving high quality. To validate this, we constructed a hierarchical template for the Paleontology application used by Shanan et al. [17]. We found that when using the ratio semantic, we were able to get an F1 score of 0.86 with precision of 0.96. On the same task, this quality is actually higher than professional human volunteers [17]. For comparison, the linear semantic achieved an F1 score of 0.76 and the logical achieved 0.73.

The factor graph we used in this Paleontology application is large enough that it is intractable, using exact inference, to estimate the true marginal to investigate the mixing behavior. Therefore, we chose a subgraph of a KBP system used by Shin et al. [20] that can be grounded by a hierarchical template and chose a setting of the weight such that the true marginal was 0.5 for all variables. We then ran Gibbs sampling on this subgraph and report the average error of the marginal estimation in Figure 5(b). Our results illustrate the effect of changing the semantic on a more complicated model from a real application, and show similar behavior to our simple voting example.

# 5   Conclusion

This paper showed that for a class of factor graph templates, hierarchical templates, Gibbs sampling mixes in polynomial time. It also introduced the graph property hierarchy width, and showed that for graphs of bounded factor weight and hierarchy width, Gibbs sampling converges rapidly. These results may aid in better understanding the behavior of Gibbs sampling for both template and general factor graphs.

**Acknowledgments**

Thanks to Stefano Ermon and Percy Liang for helpful conversations.

The authors acknowledge the support of: DARPA FA8750-12-2-0335; NSF IIS-1247701; NSF CCF-1111943; DOE 108845; NSF CCF-1337375; DARPA FA8750-13-2-0039; NSF IIS-1353606; ONR N000141210041 and N000141310129; NIH U54EB020405; Oracle; NVIDIA; Huawei; SAP Labs; Sloan Research Fellowship; Moore Foundation; American Family Insurance; Google; and Toshiba.

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
