[Supplementary Material]

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

Errors of Marginal Estimates for Synthetic Ising Model

Max Error of Marginal Estimate for KBP Dataset

(a) Error of marginal estimates for synthetic Ising model after $10^5$ samples.

(b) Maximum error marginal estimates for KBP dataset after some number of samples.

Figure 5: Experiments illustrate how convergence is affected by hierarchy width and semantics.

ranged from the star graph (like in Figure 1(a)) to the path graph; and each had different hierarchy width. For each graph, we were able to calculate the exact true marginal of each variable because of the small tree-width. We then ran Gibbs sampling on each graph, and calculated the error of the marginal estimate of a single arbitrarily-chosen query variable. Figure 5(a) shows the result with different weights and hierarchy width. It shows that, even for tree graphs with the same number of nodes and edges, the mixing time can still vary depending on the hierarchy width of the model.

**Real-World Applications** We observed that the hierarchical templates that we focus on in this work appear frequently in real applications. For example, all five knowledge base population (KBP) systems illustrated by Shin et al. [20] contain subgraphs that are grounded by hierarchical templates. Moreover, sometimes a factor graph is solely grounded by hierarchical templates, and thus provably mixes rapidly by our theorem while achieving high quality. To validate this, we constructed a hierarchical template for the Paleontology application used by Shanan et al. [17]. We found that when using the ratio semantic, we were able to get an F1 score of 0.86 with precision of 0.96. On the same task, this quality is actually higher than professional human volunteers [17]. For comparison, the linear semantic achieved an F1 score of 0.76 and the logical achieved 0.73.

The factor graph we used in this Paleontology application is large enough that it is intractable, using exact inference, to estimate the true marginal to investigate the mixing behavior. Therefore, we chose a subgraph of a KBP system used by Shin et al. [20] that can be grounded by a hierarchical template and chose a setting of the weight such that the true marginal was 0.5 for all variables. We then ran Gibbs sampling on this subgraph and report the average error of the marginal estimation in Figure 5(b). Our results illustrate the effect of changing the semantic on a more complicated model from a real application, and show similar behavior to our simple voting example.

# 5 Conclusion

This paper showed that for a class of factor graph templates, hierarchical templates, Gibbs sampling mixes in polynomial time. It also introduced the graph property hierarchy width, and showed that for graphs of bounded factor weight and hierarchy width, Gibbs sampling converges rapidly. These results may aid in better understanding the behavior of Gibbs sampling for both template and general factor graphs.

**Acknowledgments**

Thanks to Stefano Ermon and Percy Liang for helpful conversations.

The authors acknowledge the support of: DARPA FA8750-12-2-0335; NSF IIS-1247701; NSF CCF-1111943; DOE 108845; NSF CCF-1337375; DARPA FA8750-13-2-0039; NSF IIS-1353606; ONR N000141210041 and N000141310129; NIH U54EB020405; Oracle; NVIDIA; Huawei; SAP Labs; Sloan Research Fellowship; Moore Foundation; American Family Insurance; Google; and Toshiba.

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

# A  Proof of Voting Program Rates

Here, we prove the convergence rates stated in Theorem 1. The strategy for the upper bound proofs involves constructing a *coupling* between the Gibbs sampler and another process that attains the equilibrium distribution at each step. First, we restate the definition of mixing time in terms of the *total variation distance*, a quantity which we will use in the proofs in this section.

**Definition 7** (Total Variation Distance). The *total variation distance* [10, p. 48] is a distance metric between two probability measures $\mu$ and $\nu$ over probability space $\Omega$ defined as

$$\|\mu - \nu\|_{\text{TV}} = \max_{A \subset \Omega} |\mu(A) - \nu(A)|,$$

that is, the maximum difference between the probabilities that $\mu$ and $\nu$ assign to a single event.

**Definition 8** (Mixing Time). The *mixing time* of a Markov chain is the first time $t$ at which the estimated distribution $\mu_t$ is within total variation distance $\frac{1}{4}$ of the true distribution [10, p. 55]. That is,

$$t_{\text{mix}} = \min \left\{ t : \|\mu_t - \pi\|_{\text{TV}} \leq \frac{1}{4} \right\}.$$

Next, we define a coupling [30].

**Definition 9** (Coupling). A coupling of two random variables $X$ and $X'$ defined on some separate probability spaces $\mathbb{P}$ and $\mathbb{P}'$ is any new probability space $\hat{\mathbb{P}}$ over which there are two random variables $\hat{X}$ and $\hat{X}'$ such that $X$ has the same distribution as $\hat{X}$ and $X'$ has the same distribution as $\hat{X}'$.

Given a coupling of two Markov processes $X_k$ and $X'_k$ with the same transition matrix, the *coupling time* is defined as the first time $T$ when $\hat{X}_k = \hat{X}'_k$. The following theorem lets us bound the total variance distance in terms of the coupling time.

**Theorem 3** (Theorem 5.2 in Levin et al. [10]). *For any coupling $(\hat{X}_k, \hat{X}'_k)$ with coupling time $T$, if we let $\nu_k$ and $\nu'_k$ denote the distributions of $X_k$ and $X'_k$ respectively, then*

$$\|\nu_k - \nu'_k\|_{\text{tv}} \leq P(T > k).$$

All of the coupling examples in this section use a *correlated flip coupler*, which consists of two Gibbs samplers $\hat{X}$ and $\hat{X}'$, each of which is running with the same random inputs. Specifically, both samplers choose to sample the same variable at each timestep. Then, if we define $p$ as the probability of sampling 1 for $\hat{X}$, and $p'$ similarly, it assigns both variables to 1 with probability $\min(p, p')$, assigns both variables to 0 with probability $\min(1 - p, 1 - p')$, and assigns different values with probability $|p - p'|$. If we initialize $\hat{X}$ with an arbitrary distribution $\nu$ and $\hat{X}'$ with the stationary distribution $\pi$, it is trivial to check that $\hat{X}_k$ has distribution $P^k \nu$, and $\hat{X}'_k$ always has distribution $\pi$. Applying Theorem 3 results in

$$\left\| P^k \nu - \pi \right\|_{\text{tv}} \leq P(T > k).$$

Now, we prove the bounds stated in Theorem 1 as a collection of four statements about the upper and lower bounds of the mixing time.

**Statement 5** (UB for Voting: Logical and Ratio). *For the voting example, assume that all the weights on the variables are bounded by $|w| \leq M$, $|w_{T_x}| \leq M$ and $|w_{F_x}| \leq M$. Then for either the logical or ratio semantics, for sufficiently small constant $\epsilon$,*

$$t_{\text{mix}}(\epsilon) = O(n \log(n)).$$

*Proof.* Recall that, for the voting program, the weight of a world is

$$W(q, t, f) = wqg(\|t\|_1) - wqg(\|f\|_1) + \sum_{x=1}^{n} w_{T(x)} t_x + \sum_{x=1}^{n} w_{F(x)} f_x,$$

where $q \in \{-1, 1\}$ and $\{t_x, f_x\} \subseteq \{0, 1\}$.

Consider a correlated-flip coupler running on this Markov process, producing two chains $X$ and $\bar{X}$. For either chain, if we sample a variable $T(x)$ at time $k$, and let $u$ denote the number of $T(y)$ variables for $y \neq x$ that are true, then

$$P\left(T(x)^{(k+1)} = 1\right) = \left(1 + \exp\left(wq\left(g(u) - g(u+1)\right) - w_{T(x)}\right)\right)^{-1}.$$

Since $g(x+1) - g(x) \leq 1$ for any $x$, it follows that

$$P\left(T(x)^{(k+1)} = 1\right) \geq (1 + \exp(2M))^{-1}.$$

We now define the constant

$$p_M = (1 + \exp(2M))^{-1}.$$

Also, if $u$ non-$T(x)$ variables are true in the $X$ process, and $\bar{u}$ variables are true in the $\bar{X}$ process, then the probability of the processes $X$ and $\bar{X}$ not producing the same value for $Y_i$ is

$$p_D = \left|P\left(T(x)^{(k+1)} = 1\right) - P\left(\bar{T}(x)^{(k+1)} = 1\right)\right|$$
$$= \left|\left(1 + \exp\left(wq\left(g(u) - g(u+1)\right) - w_{T(x)}\right)\right)^{-1} - \left(1 + \exp\left(w\bar{q}\left(g(\bar{u}) - g(\bar{u}+1)\right) - w_{T(x)}\right)\right)^{-1}\right|,$$

where the last statement follows from continuous differentiability of the function $h(x) = (1 + \exp(-x))^{-1}$. Notice that, if $u = \Omega(n)$, then

$$p_D = O(n^{-1})$$

for both logical and ratio semantics.

The same logic will show that, if we sample any $F(x)$, then

$$P\left(F(x)^{(k+1)} = 1\right) \geq p_M,$$

and

$$p_D = O(g(v+1) - g(v) + g(\bar{v}+1) - g(\bar{v})),$$

where $v$ and $\bar{v}$ are the number of non-$F(x)$ variables that are true in the $X$ and $\bar{X}$ processes respectively. Again as above, if $v = \Omega(n)$, then

$$p_D = O(n^{-1}).$$

Next, consider the situation if we sample $Q$. In this case,

$$P\left(Q^{(k+1)} = 1\right) = \left(1 + \exp\left(2w\left(g(\|t\|_1) - g(\|f\|_1)\right)\right)\right)^{-1}.$$

Therefore, of we let $u$ denote the number of true $T(x)$ variables and $v$ denote the number of true $F(x)$ variables, and similarly for $\bar{u}$ and $\bar{v}$, then the probability of the processes $X$ and $\bar{X}$ not producing the same value for $Q$ is

$$p_D = \left|P\left(Q^{(k+1)} = 1\right) - P\left(\bar{Q}^{(k+1)} = 1\right)\right|$$
$$= \left|\left(1 + \exp\left(2w\left(f(v) - f(u)\right)\right)\right)^{-1}. - \left(1 + \exp\left(2w\left(f(\bar{v}) - f(\bar{u})\right)\right)\right)^{-1}.\right|$$
$$= O\left(|f(v) - f(u) - f(\bar{v}) + f(\bar{u}),|\right)$$

where as before, the last statement follows from continuous differentiability of the function $g(x) = (1 + \exp(-x))^{-1}$. Furthermore, if all of $u$, $v$, $\bar{u}$, and $\bar{v}$ are $\Omega(n)$, then

$$p_D = O(1).$$

Now, assume that our correlated-flip coupler runs for $8n \log n$ steps on the Gibbs sampler. Let $E_1$ be the event that, after the first $4n \log n$ steps, each of the variables has been sampled at least once. This will have probability at least $\frac{1}{2}$ by the coupon collector's problem.

Next, let $E_2$ be the event that, after the first $4n \log n$ steps, $\min(u, v, \bar{u}, \bar{v}) \geq \frac{p_M n}{2}$ for the next $4n \log n$ steps for both samplers. After all the entries have been sampled, $u$, $v$, $\bar{u}$, and $\bar{v}$ will each

be bounded from below by a binomial random variable with parameter $p_M$ at each timestep, so from Hoeffding's inequality, the probability that this constraint will be violated by a sampler at any particular step is less than $\exp\left(-\frac{1}{2}p_M n\right)$. Therefore,

$$\mathbf{P}\left(E_2 | E_1\right) \geq \left(1 - \exp\left(-\frac{1}{2}p_M n\right)\right)^{4n \log n} = \Omega(1).$$

Let $E_3$ be the event that all the variables are resampled at least once between time $2n \log n$ and $4n \log n$. Again this event has probability at least $\frac{1}{2}$.

Finally, let $C$ be the event that coupling occurs at time $4n \log n$. Given $E_1$, $E_2$, and $E_3$, this probability is equal to the probability that each variable coupled individually the last time it was sampled. For all the $Y_i$, our analysis above showed that this probability is

$$1 - p_D = 1 - O(n^{-1}),$$

and for $Q$, this probability is

$$1 - p_D = \Omega(1).$$

Therefore,

$$\mathbf{P}\left(C | E_1, E_2, E_3\right) \geq \Omega(1)\left(1 - O(n^{-1})\right)^{2n} = \Omega(1),$$

and since

$$P\left(C\right) = \mathbf{P}\left(C | E_1, E_2, E_3\right) P\left(E_3\right) \mathbf{P}\left(E_2 | E_1\right) P\left(E_1\right),$$

and all these quantities are $\Omega(1)$, we can conclude that $\hat{\mathbb{P}}(C) = \Omega(1)$.

Therefore, this process couples with at least some constant probability $P$ independent of $n$ after $8n \log n$ steps. Since we can run this coupling argument independently an arbitrary number of times, it follows that, after $8Ln \log n$ steps, the probability of coupling will be at least $1 - (1-P)^L$. Therefore, by Theorem 3, for any initial distribution $\nu$,

$$\|P^{8Ln \log n}\nu - \pi\|_{\text{tv}} \leq \hat{\mathbb{P}}(T > 8Ln \log n) \leq (1-P)^L.$$

For any $\epsilon$, this will be less than $\epsilon$ when

$$L \geq \frac{\log(\epsilon)}{\log(1 - p_C)},$$

which occurs when $t \geq 8n \log n \frac{\log(\epsilon)}{\log(1-p_C)}$. Letting $\epsilon = \frac{1}{4}$,

$$t_{\text{mix}} = 8n \log n \frac{\log(4)}{-\log(1 - p_C)}$$

produces the desired result. $\qquad\square$

**Statement 6** (LB for Voting: Logical and Ratio). *For the voting example using either the logical or ratio semantics, a lower bound for the mixing time for sufficiently small constant values of $\epsilon$ is $\Omega(n \log n)$.*

*Proof.* At a minimum, in order to converge, we must sample all the variables. From the coupon collector's problem, this requires $\Omega(n \log n)$ time. $\qquad\square$

**Statement 7** (UB for Voting: Linear). *For the voting example, assume that all the weights on the variables are bounded by $|w| \leq M$, $\left|w_{T(x)}\right| \leq M$ and $\left|w_{F(x)}\right| \leq M$. Then for linear semantics,*

$$t_{\text{mix}} = 2^{O(n)}.$$

*Proof.* From our algebra above in the proof of Statement 5, we know that at any timestep, the probability of not coupling the sampled variable is bounded; that is

$$p_D = O(1).$$

Therefore, if we run a correlated-flip coupler for $2n \log n$ timesteps, the probability that coupling will have occurred is greater than the probability that all variables have been sampled (which is

$\Omega(1))$ times the probability that all variables coupled the last time they were sampled. Thus if $C$ is the event that coupling occurs,

$$P\left(C\right) \geq \Omega(1)\left(1 - O(1)\right)^{2n+1} = \exp(-O(n)).$$

Therefore, if we run for some $t = 2^{O(n)}$ timesteps, coupling will have occurred at least once with high probability. It follows that the mixing time is

$$t_{\mathrm{mix}} = 2^{O(n)},$$

as desired. □

**Statement 8** (LB for Voting: Linear). *For the voting example using linear semantics and bounded weights, a lower bound for the mixing time of a worst-case model is $2^{\Omega(n)}$.*

*Proof.* Consider a sampler with unary weights $w_{T(x)} = w_{F(x)} = 0$. Assume that it starts in a state where $Q = 1$, all the $T(x) = 1$, and all the $F(x) = 0$. We will show that it takes an exponential amount of time until $Q = -1$. From above, the probability of flipping $Q$ will be

$$P\left(Q^{(k+1)} = -1\right) = \left(1 + \exp\left(2w\left(\|t\|_1\right) - \|f\|_1\right)\right)\right)^{-1}.$$

Meanwhile, the probability to flip any $T(x)$ while $Q = 1$ is

$$P\left(T(x)^{(k+1)} = 0\right) = \left(1 + \exp(w)\right)^{-1} = p,$$

for constant $p$, and the probability of flipping any $F(x)$ is similarly

$$P\left(F(x)^{(k+1)} = 1\right) = p.$$

Now, consider the following events which could happen at any timestep. While $Q = 1$, let $E_T$ be the event that $\|t\|_1 \leq (1 - 2p)n$, and let $E_F$ be the event that $\|f\|_1 \geq 2pn$. Since $\|t\|_1$ and $\|f\|_1$ are both bounded by binomial random variables with parameters $1 - p$ and $p$ respectively, Hoeffding's inequality states that, at any timestep,

$$P\left(E_T\right) = P\left(\|t\|_1 \leq (1 - 2p)n\right) \leq \exp(-2p^2 n),$$

and similarly

$$P\left(E_F\right) = P\left(\|f\|_1 \geq 2pn\right) \leq \exp(-2p^2 n).$$

Now, while these bounds are satisfied, let $E_Q$ be the event that $Q = -1$. This will be bounded by

$$P\left(E_Q\right) = \left(1 + \exp\left(2w(1 - 4p)n\right)\right)^{-1} \leq \exp(-2w(1 - 4p)n).$$

It follows that at any timestep,

$$P\left(E_T \vee E_F \vee E_Q\right) = \exp(-O(n)),$$

so, at any timestep $k$,

$$P\left(Q^{(k)} = -1\right) = k\exp(-O(n)).$$

However, by symmetry, under the stationary distribution $\pi$, this probability must be $\frac{1}{2}$. Therefore, the total variation distance is bounded by

$$\|P_t \nu - \pi\|_{\mathrm{tv}} \geq \frac{1}{2} - t\exp(-O(n)).$$

So, for convergence to less than $\epsilon = \frac{1}{4}$, for example, we must require at least $2^{O(a)}$ steps. This proves the statement. □

## B  Proof of Theorem 2

In this section, we prove the main result of the paper, Theorem 2. First, we state some basic lemmas we will need for the proof in Section B.1. Then, we prove the main result inductively in Section B.2. Finally, we prove the lemmas in Section B.3.

### B.1 Statement of Lemmas

Note that some of these lemmas are restated from the body of the paper.

**Definition 10** (Absolute Spectral Gap). Let $P$ be the transition matrix of a Markov process. Since it is a Markov process, one of its eigenvalues, the dominant eigenvalue, must be 1. The *absolute spectral gap* of the Markov process is the value

$$\gamma = 1 - \max_{\lambda} |\lambda|,$$

where the maximum is taken over all non-dominant eigenvalues of $P$.

**Lemma 5** (Disconnected Case). *Let $G$ be a disconnected factor graph with $n$ variables and $m$ connected components $G_1, G_2, \ldots, G_m$ with $n_1, n_2, \ldots n_m$ variables, respectively. Then,*

$$\gamma(G) \geq \min_{i \leq m} \frac{n_i}{n} \gamma(G_i).$$

**Lemma 4** (Connected Case). *Let $G$ and $\bar{G}$ be two factor graphs with maximum factor weight $M$, which differ only inasmuch as $G$ contains a single additional factor $\phi^*$. Then,*

$$\gamma(G) \geq \gamma(\bar{G}) \exp(-3M).$$

**Lemma 6** (Base Case). *Let $G$ be a factor graph with one variable and no factors. The absolute spectral gap of Gibbs sampling running on $G$ will be $\gamma(G) = 1$.*

**Lemma 8.** *Let $P$ be the transition matrix of a reversible, irreducible Markov chain with state space $\Omega$ and absolute spectral gap $\gamma$, and let $\pi_{\min} = \min_{x \in \pi} \pi(x)$. Then*

$$t_{\mathrm{mix}}(\epsilon) \leq -\log(\epsilon \pi_{\min}) \frac{1}{\gamma}.$$

*Proof.* This is Theorem 12.3 from Markov Chains and Mixing Times [10, p. 155], and a complete proof can be found in that book. $\square$

### B.2 Main Proof

We achieve the proof of Theorem 2 in two steps. First, we inductively bound the absolute spectral gap of Gibbs sampling on factor graphs in terms of the hierarchy width. Then, we use the bound on the spectral gap to bound the mixing time.

**Lemma 9.** *If $G$ is a factor graph with $n$ variables, maximum factor weight $M$ and hierarchy width $h$, then Gibbs sampling running on $G$ will have absolute spectral gap $\gamma$, where*

$$\gamma \geq \frac{1}{n} \exp(-3hM).$$

*Proof.* We will prove this result by multiple induction on the number of variables and number of factors of $G$. In what follows, we assume that the statement holds for all graphs with either fewer variables and no more factors than $G$, or fewer factors and no more variables than $G$.

Consider the connectedness of $G$. There are three possibilities:

1. $G$ has one variable and no factors.
2. $G$ is connected and has at least one factor.
3. $G$ is disconnected.

We consider these cases separately.

**Case 1**   If $G$ has no factors and one variable, then by Lemma 6, its spectral gap will be

$$\gamma = 1.$$

Also, if $G$ has no factors, by (4), its hierarchy width is $h = 0$. Therefore,

$$\gamma = \frac{1}{1} \exp(0)$$
$$= \frac{1}{n} \exp(-3hM),$$

which is the desired result.

**Case 2** Next, we consider the case where $G$ is connected and has at least one factor. In this case, by (2), its hierarchy width is

$$\mathsf{hw}(G) = 1 + \min_{e \in E} \mathsf{hw}(\langle N, E - \{e\} \rangle).$$

Let $e$ be an edge that minimizes this quantity, and let $\bar{G}$ denote the factor graph that results from removing the corresponding factor from $G$. Clearly,

$$\mathsf{hw}(G) = 1 + \mathsf{hw}(\bar{G}).$$

Since $G$ and $\bar{G}$ differ by only one factor, by Lemma 4,

$$\gamma \geq \bar{\gamma} \exp\left(-3M\right).$$

Furthermore, since $\bar{G}$ has fewer factors than $G$, by the inductive hypothesis,

$$\bar{\gamma} \geq \frac{1}{n} \exp\left(-3M\mathsf{hw}(\bar{G})\right).$$

Therefore,

$$\begin{aligned}
\gamma &\geq \bar{\gamma} \exp\left(-M\right) \\
&\geq \frac{1}{n} \exp\left(-3M\mathsf{hw}(\bar{G})\right) \exp\left(-3M\right) \\
&= \frac{1}{n} \exp\left(-3M\left(1 + \mathsf{hw}(\bar{G})\right)\right) \\
&= \frac{1}{n} \exp\left(-3M\mathsf{hw}(G)\right),
\end{aligned}$$

which is the desired result.

**Case 3** Finally, consider the case where $G$ is disconnected. Let $G_1, G_2, \ldots, G_m$ be the connected components of $G$ for some $m \geq 2$. In this case, by (3), its hierarchy width is

$$\mathsf{hw}(G) = \max_i \mathsf{hw}(G_i).$$

By Lemma 5,

$$\gamma = \min_{i \leq m} \frac{n_i}{n} \gamma_i,$$

where $n_i$ are the sizes of the $G_i$ and $\gamma_i$ are their absolute spectral gaps. We further know that each of the $G_i$ has fewer nodes than $G$, so by the inductive hypothesis,

$$\gamma_i \geq \frac{1}{n_i} \exp\left(-3M\mathsf{hw}(G_i)\right).$$

Therefore,

$$\begin{aligned}
\gamma &= \min_{i \leq m} \frac{n_i}{n} \gamma_i \\
&\geq \min_{i \leq m} \frac{n_i}{n} \frac{1}{n_i} \exp\left(-3M\mathsf{hw}(G_i)\right) \\
&= \min_{i \leq m} \frac{1}{n} \exp\left(-3M\mathsf{hw}(G_i)\right) \\
&= \frac{1}{n} \exp\left(-3M \max_{i \leq m} \mathsf{hw}(G_i)\right) \\
&= \frac{1}{n} \exp\left(-3M\mathsf{hw}(G)\right),
\end{aligned}$$

which is the desired result.

Since the statement holds in all three cases, by induction it will hold for all factor graphs. This completes the proof. $\qquad\square$

Now, we restate and prove the main theorem.

**Theorem 2** (Polynomial Mixing Time). *If $G$ is a factor graph with $n$ variables, at most $s$ states per variable, $e$ factors, maximum factor weight $M$, and hierarchy width $h$, then*

$$t_{\text{mix}} \leq (\log(4) + n\log(s) + eM)\, n \exp(3hM).$$

*In particular, if $e$ is polynomial in $n$, the number of values for each variable is bounded, and $hM = O(\log n)$, then $t_{\text{mix}}(\epsilon) = O(n^{O(1)})$.*

*Proof.* From Lemma 8, we have that

$$t_{\text{mix}}(\epsilon) \leq -\log\left(\epsilon \pi_{\min}\right) \frac{1}{\gamma}.$$

For our factor graph $G$,

$$\pi_{\min} = \min_I \pi(I)$$
$$= \frac{\min_I \exp(W(I))}{\sum_J \exp(W(J))}.$$

Since there are only at most $s^n$ worlds, it follows that

$$\pi_{\min} \geq \frac{\min_I \exp(W(I))}{s^n \max_J \exp(W(J))}$$
$$= s^{-n} \exp\left(\min_I W(I) - \max_J W(J)\right).$$

Expanding the world weight in terms of the factors,

$$\pi_{\min} \geq s^{-n} \exp\left(\min_I \sum_{\phi \in \Phi} \phi(I) - \max_J \sum_{\phi \in \Phi} \phi(J)\right)$$
$$\geq s^{-n} \exp\left(\sum_{\phi \in \Phi}\left(\min_I \phi(I) - \max_J \phi(J)\right)\right).$$

Applying our maximum factor weight bound,

$$\pi_{\min} \geq s^{-n} \exp\left(-\sum_{\phi \in \Phi} M\right)$$
$$= s^{-n} \exp\left(-eM\right).$$

Substituting this into the expression from Lemma 8 produces

$$t_{\text{mix}}(\epsilon) \leq -\log\left(\frac{\epsilon}{s^n} \exp\left(-eM\right)\right) \frac{1}{\gamma}$$
$$= (n\log(s) + eM - \log(\epsilon)) \frac{1}{\gamma},$$

and substituting the result from Lemma 9 gives

$$t_{\text{mix}}(\epsilon) \leq (n\log(s) + eM - \log(\epsilon))\, n \exp(3hM).$$

Substituting $\epsilon = \frac{1}{4}$ gives the desired result. $\qquad\square$

## B.3 Proofs of Lemmas

In this statement, we will restate and prove the lemmas stated above in Section B.1.

**Lemma 5** (Disconnected Case). *Let $G$ be a disconnected factor graph with $n$ variables and $m$ connected components $G_1, G_2, \ldots, G_m$ with $n_1, n_2, \ldots n_m$ variables, respectively. Then,*

$$\gamma(G) \geq \min_{i \leq m} \frac{n_i}{n} \gamma(G_i).$$

*Proof.* Lemma 5 follows directly from Corollary 12.12 in Markov Chains and Mixing Times [10, p. 161]. For completeness, we restate the proof here.

Since $G$ is disconnected, Gibbs sampling on $G$ is equivalent to running $m$ independent Gibbs samplers on the $G_i$, where at each timestep, the Gibbs sampler of $G_i$ is updated if a variable in $G_i$ is chosen; this occurs with probability $\frac{n_i}{n}$. Therefore, if we let $P_i$ denote the Markov transition matrix of Gibbs sampling on $G_i$, we can write the transition matrix $P$ of Gibbs sampling on $G$ as

$$P(x, y) = \sum_{i=1}^{m} \frac{n_i}{n} P_i(x_i, y_i) \prod_{j \neq i} I(x_j, y_j),$$

where $x$ and $y$ are worlds on $G$, where $x_i$ and $y_i$ are subsets of the variables of $x$ and $y$ respectively that correspond to the graph $G_i$, and $I$ is the identity matrix ($I(x, y) = 1$ if $x = y$, and $I(x, y) = 0$ otherwise).

Now, we, for each $i$, we let $f_i$ be some eigenvector of $P_i$ with eigenvalue $\lambda_i$, and define

$$f(x) = \prod_{i=1}^{m} f_i(x_i).$$

It follows that

$$(Pf)(y) = \sum_{x} f(x) P(x, y)$$

$$= \sum_{x} f(x) \sum_{i=1}^{m} \frac{n_i}{n} P_i(x_i, y_i) \prod_{j \neq i} I(x_j, y_j)$$

$$= \sum_{x} \sum_{i=1}^{m} \frac{n_i}{n} f_i(x_i) P_i(x_i, y_i) \prod_{j \neq i} f_j(x_j) I(x_j, y_j)$$

$$= \sum_{i=1}^{m} \frac{n_i}{n} \sum_{x_i} f_i(x_i) P_i(x_i, y_i) \prod_{j \neq i} \sum_{x_j} f_j(x_j) I(x_j, y_j)$$

$$= \sum_{i=1}^{m} \frac{n_i}{n} \lambda_i f_i(y_i) \prod_{j \neq i} f_j(y_j)$$

$$= f(y) \sum_{i=1}^{m} \frac{n_i}{n} \lambda_i;$$

if follows that $f$ is an eigenvector of $P$ with eigenvalue

$$\lambda = \sum_{i=1}^{m} \frac{n_i}{n} \lambda_i. \tag{9}$$

Furthermore, it is clear that such eigenvectors will form a basis for the space of possible distributions on the worlds of $G$. Therefore, all eigenvalues of $P$ will be of the form (9), and so

$$|\lambda| \leq \sum_{i=1}^{m} \frac{n_i}{n} |\lambda_i|.$$

Therefore,

$$\gamma = \min_{\lambda \neq 1} (1 - |\lambda|)$$

$$\geq \min_{\lambda \neq 1} \sum_{i=1}^{m} \frac{n_i}{n} (1 - |\lambda_i|)$$

$$\geq \min_i \min_{\lambda_i \neq 1} \frac{n_i}{n} (1 - |\lambda_i|)$$

$$= \min_i \frac{n_i}{n} \gamma_i,$$

which is the desired result. $\square$

To prove Lemma 4, we will need to first prove two other lemmas.

**Lemma 10.** *Let $G$ and $\bar{G}$ be two factor graphs, which differ only inasmuch as $G$ contains a single additional factor $\phi^*$ with maximum factor weight $M$. Then, if $\pi$ and $\bar{\pi}$ denote the distributions on worlds for these graphs,*

$$\exp(-M)\bar{\pi}(x) \leq \pi(x) \leq \exp(M)\bar{\pi}(x).$$

*Proof.* From the definition of the distribution function,

$$\pi(x) = \frac{\exp(W(x))}{\sum_{z \in \Omega} \exp(W(z))}$$

$$= \frac{\exp(\bar{W}(x) + \phi^*(x))}{\sum_{z \in \Omega} \exp(\bar{W}(z) + \phi^*(z))}$$

$$\geq \frac{\exp(\bar{W}(x) + \min_I \phi^*(I))}{\sum_{z \in \Omega} \exp(\bar{W}(z) + \max_I \phi^*(I))}$$

$$= \frac{\exp(\bar{W}(x))}{\sum_{z \in \Omega} \exp(\bar{W}(z))} \exp\left(\min_I \phi^*(I) - \max_I \phi^*(I)\right)$$

$$= \exp(-M)\bar{\pi}(x).$$

In the other direction,

$$\pi(x) = \frac{\exp(\bar{W}(x) + \phi^*(x))}{\sum_{z \in \Omega} \exp(\bar{W}(z) + \phi^*(z))}$$

$$\leq \frac{\exp(\bar{W}(x) + \max_I \phi^*(I))}{\sum_{z \in \Omega} \exp(\bar{W}(z) + \min_I \phi^*(I))}$$

$$= \frac{\exp(\bar{W}(x))}{\sum_{z \in \Omega} \exp(\bar{W}(z))} \exp\left(\max_I \phi^*(I) - \min_I \phi^*(I)\right)$$

$$= \exp(M)\bar{\pi}(x).$$

This proves the lemma. $\square$

**Lemma 11.** *Let $G$ and $\bar{G}$ be two factor graphs, which differ only inasmuch as $G$ contains a single additional factor $\phi^*$ with maximum factor weight $M$. Then, if $P$ and $\bar{P}$ denote the transition matrices for Gibbs sampling on these graphs, for all $x \neq y$,*

$$\exp(-M)\bar{P}(x, y) \leq P(x, y) \leq \exp(M)\bar{P}(x, y).$$

*Proof.* For Gibbs sampling on $G$, we know that, if worlds $x$ and $y$ differ in the value of only one variable, and $L$ is the set of worlds that only differ from $x$ and $y$ in the same variable, then

$$P(x, y) = \frac{1}{n} \frac{\exp(W(y))}{\sum_{l \in L} \exp(W(l))}$$

If $\bar{P}$ is the transition matrix for Gibbs sampling on $\bar{G}$, then the same argument will show that

$$\bar{P}(x,y) = \frac{1}{n} \frac{\exp(\bar{W}(y))}{\sum_{l \in L} \exp(\bar{W}(l))}.$$

Therefore,

$$\begin{aligned}
P(x,y) &= \frac{1}{n} \frac{\exp(\bar{W}(y) + \phi^*(y))}{\sum_{l \in L} \exp(\bar{W}(l) + \phi^*(y))} \\
&\geq \frac{1}{n} \frac{\exp(\bar{W}(y) + \min_I \phi^*(I))}{\sum_{l \in L} \exp(\bar{W}(l) + \max_I \phi^*(I))} \\
&= \frac{1}{n} \frac{\exp(\bar{W}(y))}{\sum_{l \in L} \exp(\bar{W}(l))} \exp\left(\min_I \phi^*(I) - \max_I \phi^*(I)\right) \\
&= \exp(-M)\bar{P}(x,y).
\end{aligned}$$

Similarly,

$$\begin{aligned}
P(x,y) &= \frac{1}{n} \frac{\exp(\bar{W}(y) + \phi^*(y))}{\sum_{l \in L} \exp(\bar{W}(l) + \phi^*(y))} \\
&\leq \frac{1}{n} \frac{\exp(\bar{W}(y) + \max_I \phi^*(I))}{\sum_{l \in L} \exp(\bar{W}(l) + \min_I \phi^*(I))} \\
&= \frac{1}{n} \frac{\exp(\bar{W}(y))}{\sum_{l \in L} \exp(\bar{W}(l))} \exp\left(\max_I \phi^*(I) - \min_I \phi^*(I)\right) \\
&= \exp(M)\bar{P}(x,y).
\end{aligned}$$

On the other hand, if $x$ and $y$ differ in more than one variable, then

$$P(x,y) = \bar{P}(x,y) = 0.$$

Therefore, the lemma holds for any $x \neq y$. $\qquad\square$

Now, we restate and prove Lemma 4.

**Lemma 4** (Connected Case). *Let $G$ and $\bar{G}$ be two factor graphs with maximum factor weight $M$, which differ only inasmuch as $G$ contains a single additional factor $\phi^*$. Then,*

$$\gamma(G) \geq \gamma(\bar{G}) \exp\left(-3M\right).$$

*Proof.* The proof is similar to proofs using the Dirichlet form to bound the spectral gap [10, p. 181].

Recall that Gibbs sampling is a reversible Markov chain. That is, if $P$ is the transition matrix associated with Gibbs sampling on $G$, and $\pi$ is its stationary distribution,

$$\pi(x)P(x,y) = \pi(y)P(y,x).$$

If we let $\Pi$ be the diagonal matrix such that $\Pi(x,x) = \pi(x)$, then we can write this as

$$P\Pi = \Pi P^T;$$

that is, $A = P\Pi$ is a symmetric matrix. Now, consider the form

$$\alpha(f) = \frac{1}{2} \sum_{x,y} \left(f(x) - f(y)\right)^2 \pi(x)P(x,y).$$

Expanding this produces

$$\alpha(f) = \frac{1}{2} \sum_{x,y} f(x)^2 \pi(x)P(x,y) - \sum_{x,y} f(x)f(y)\pi(x)P(x,y) + \frac{1}{2} \sum_{x,y} f(y)^2 \pi(x)P(x,y).$$

By reversibility of the chain,

$$\alpha(f) = \frac{1}{2}\sum_{x,y} f(x)^2 \pi(x)P(x,y) - \sum_{x,y} f(x)f(y)\pi(x)P(x,y) + \frac{1}{2}\sum_{x,y} f(y)^2 \pi(y)P(y,x)$$

$$= \sum_{x,y} f(x)^2 \pi(x)P(x,y) - \sum_{x,y} f(x)f(y)\pi(x)P(x,y)$$

$$= \sum_{x} f(x)^2 \pi(x) \sum_{y} P(x,y) - \sum_{x,y} f(x)f(y)\pi(x)P(x,y)$$

$$= \sum_{x} \pi(x)f(x)^2 - \sum_{x,y} f(x)f(y)A(x,y)$$

$$= f^T \Pi f - f^T A f$$

$$= f^T (I - P)\Pi f.$$

Similarly, if we define

$$\beta(f) = \frac{1}{2}\sum_{x,y} \left( f(x) + f(y) \right)^2 \pi(x)P(x,y),$$

then the same logic will show that

$$\beta(f) = f^T (I + P)\Pi f.$$

We define $\bar{A}$, $\bar{\alpha}$ and $\bar{\beta}$ similarly for Gibbs sampling on $\bar{G}$.

Now, by Lemmas 10 and 11,

$$\alpha(f) = \frac{1}{2}\sum_{x,y} \left( f(x) - f(y) \right)^2 \pi(x)P(x,y)$$

$$\geq \frac{1}{2}\sum_{x,y} \left( f(x) - f(y) \right)^2 \exp(-2M)\bar{\pi}(x)\bar{P}(x,y)$$

$$= \exp(-2M)\bar{\alpha}(f).$$

It follows that

$$f^T (I - P)\Pi f \geq \exp(-2M) f^T (I - \bar{P})\bar{\Pi} f.$$

We also notice that, by Lemma 10,

$$f^T \left( \Pi - \pi\pi^T \right) f = f^T \Pi f - (\pi^T f)^2$$

$$= \sum_{x} f(x)^2 \pi(x) - \left( \sum_{y} f(y)\pi(y) \right)^2$$

$$= \sum_{x} \left( f(x) - \sum_{y} f(y)\pi(y) \right)^2 \pi(x)$$

$$\leq \sum_{x} \left( f(x) - \sum_{y} f(y)\bar{\pi}(y) \right)^2 \pi(x)$$

$$\leq \exp(M) \sum_{x} \left( f(x) - \sum_{y} f(y)\bar{\pi}(y) \right)^2 \bar{\pi}(x)$$

$$= \exp(M) \left( \sum_{x} f(x)^2 \bar{\pi}(x) - \left( \sum_{y} f(y)\bar{\pi}(y) \right)^2 \right)$$

$$= \exp(M) f^T \left( \bar{\Pi} - \bar{\pi}\bar{\pi}^T \right) f$$

Therefore,

$$\frac{f^T (I - P)\Pi f}{f^T \left( \Pi - \pi\pi^T \right) f} \geq \exp(-3M) \frac{f^T (I - \bar{P})\bar{\Pi} f}{f^T \left( \bar{\Pi} - \bar{\pi}\bar{\pi}^T \right) f}. \tag{10}$$

Now, let's explore the quantity

$$\min_f \frac{f^T(I - P)\Pi f}{f^T\left(\Pi - \pi\pi^T\right)f}. \tag{11}$$

First, we notice that

$$(I - P)\Pi\mathbf{1} = (I - P)\pi = \pi - \pi = 0,$$

so $\mathbf{1}$ is an eigenvector of $(I - P)\Pi$ with eigenvalue 0. Furthermore,

$$\left(\Pi - \pi\pi^T\right)\mathbf{1} = \left(\pi - \pi\pi^T\mathbf{1}\right) = \pi - \pi = 0,$$

so $\mathbf{1}$ is also an eigenvector of $(I-P)\Pi$ with eigenvalue 0. It follows that (11) is invariant to additions of the vector $\mathbf{1}$ to $f$, and in particular we can therefore choose to minimize over only those vectors for which $\pi^T f = 0$. Therefore,

$$\min_f \frac{f^T(I - P)\Pi f}{f^T\left(\Pi - \pi\pi^T\right)f} = \min_{\pi^T g=0} \frac{g^T(I - P)\Pi g}{g^T\Pi g}$$

$$= \min_{\pi^T g=0} \frac{g^T\Pi^{\frac{1}{2}}\Pi^{-\frac{1}{2}}(I - P)\Pi^{\frac{1}{2}}\Pi^{\frac{1}{2}}g}{g^T\Pi g}.$$

If we let $h = \Pi^{\frac{1}{2}}g$, then

$$\min_f \frac{f^T(I - P)\Pi f}{f^T\left(\Pi - \pi\pi^T\right)f} = \min_{\mathbf{1}^T\Pi^{\frac{1}{2}}h=0} \frac{h^T\Pi^{-\frac{1}{2}}(I - P)\Pi^{\frac{1}{2}}h}{h^T h}.$$

Notice that $\Pi^{-\frac{1}{2}}(I - P)\Pi^{\frac{1}{2}}$ is similar to $I - P$, and so it must have the same eigenvalues. Since it is symmetric, it has an orthogonal complement of eigenvectors, and the eigenvector that corresponds to $\lambda = 0$ is

$$\left(\Pi^{-\frac{1}{2}}(I - P)\Pi^{\frac{1}{2}}\right)\left(\Pi^{\frac{1}{2}}\mathbf{1}\right) = \Pi^{-\frac{1}{2}}(I - P)\pi = \Pi^{-\frac{1}{2}}(\pi - \pi) = 0.$$

The expression above is minimizing over all vectors orthogonal to this eigenvector; it follows that the minimum will be the second smallest eigenvalue of $I - P$. This eigenvalue will be $1 - \lambda_2$, where $\lambda_2$ is the second-largest eigenvalue of $P$. So,

$$\min_f \frac{f^T(I - P)\Pi f}{f^T\left(\Pi - \pi\pi^T\right)f} = 1 - \lambda_2.$$

The same argument will show that

$$\min_f \frac{f^T(I - \bar{P})\bar{\Pi}f}{f^T\left(\bar{\Pi} - \bar{\pi}\bar{\pi}^T\right)f} = 1 - \bar{\lambda}_2.$$

Therefore, minimizing both sides in (10) produces

$$1 - \lambda_2 \geq \exp(-3M)\left(1 - \bar{\lambda}_2\right).$$

And applying the definition of absolute spectral gap,

$$1 - \lambda_2 \geq \exp(-3M)\bar{\gamma}$$

Now, a similar argument to the above will show that

$$\frac{f^T(I + P)\Pi f}{f^T\Pi f} \geq \exp(-3M)\frac{f^T(I + \bar{P})\bar{\Pi}f}{f^T\bar{\Pi}f}.$$

And we will be able to conclude that

$$\min_f \frac{f^T(I + P)\Pi f}{f^T\Pi f} = 1 - \lambda_n,$$

where $\lambda_n$ is the (algebraically) smallest eigenvalue of $P$, and similarly for $\bar{\lambda}_n$. Therefore, we can conclude that

$$1 + \lambda_n \geq \exp(-3M)\left(1 + \bar{\lambda}_n\right),$$

and from the definition of absolute spectral gap,

$$1 + \lambda_n \geq \exp(-3M)\bar{\gamma}.$$

Therefore,

$$\gamma \geq \min(1 - \lambda_2, 1 + \lambda_n) \geq \exp(-3M)\bar{\gamma}.$$

This proves the lemma. $\qquad\square$

Next, we restate and prove Lemma 6

**Lemma 6** (Base Case). *Let $G$ be a factor graph with one variable and no factors. The absolute spectral gap of Gibbs sampling running on $G$ will be $\gamma(G) = 1$.*

*Proof.* Gibbs sampling on a factor graph with one variable and no factors will have transition matrix

$$P = \pi \mathbf{1}^T,$$

where $\pi$ is the stationary distribution, and $\mathbf{1}$ is the vector of all 1s. That is, the process achieves the stationary distribution in a single step. The only eigenvalues of this matrix are 0 and 1, so the absolute spectral gap will be 1, as desired. □

## C  Proofs of Other Results

In this section, we will prove the other results about hierarchy decomposition stated in Section 2.3. First, we express it in terms of a hypergraph decomposition; this is how hypertree width is defined so it is useful for comparison.

**Definition 11** (Hierarchy Decomposition). A hierarchy decomposition of a hypergraph $G = \langle N, E \rangle$ is a rooted tree $T$ where each node $v$ of $T$ is labeled with a set $\chi(v)$ of edges of $G$, that satisfies the following conditions: (1) for each edge $e \in E$, there is some node $v$ of $T$ such that $e \in \chi(v)$; (2) if for two nodes $u$ and $v$ of $T$ and for some edge $e \in E$, both $e \in \chi(u)$ and $e \in \chi(v)$, then for all nodes $w$ on the (unique) path between $u$ and $v$ in $T$, $e \in \chi(w)$; (3) for every pair of edges $e \in E$ and $f \in E$, if $e \cap f \neq \emptyset$, then there exists a node $v$ of $T$ such that $\{e, f\} \subseteq \chi(v)$; and (4) if node $u$ is the parent of node $v$ in $T$, then $u \subseteq v$.

The width of a hierarchy decomposition is the size of the largest node in $T$, that is, $\max_v |\chi(v)|$.

**Statement 9.** *The* hierarchy width *of a hypergraph $G$ is equal to the minimum width of a hierarchy decomposition of $G$.*

In this section, we denote the *width* of a hierarchy decomposition $T$ as $\mathsf{w}(T)$.

**Lemma 12.** *Let $T$ be a hierarchy decomposition of a hypergraph $G$. Let $u$ be a node of $T$ that has exactly one child $v$, and let $e$ be the edge connecting $u$ and $v$. Let $\bar{T}$ be the graph minor of $T$ that is formed by removing $u$ from $T$ and, if applicable, connecting $v$ with the parent of $u$; assume that the nodes $\bar{T}$ have the same labeling as the corresponding nodes in $T$. Then $\bar{T}$ is a hierarchy decomposition of $G$, and*

$$\mathsf{w}(T) = \mathsf{w}(\bar{T}).$$

*Proof.* We validate the conditions of the definition of hierarchy decomposition individually.

1. Since $T$ is a hierarchy decomposition of $G$, by condition (1), for any hyperedge $e$ of $G$, there exists a node $x$ of $T$ such that $e \in \chi_T(x)$. If $x \neq u$, then $e \in \chi_{\bar{T}}(x)$. Otherwise, by condition (4) of $T$, it will hold that $\chi_T(u) \subseteq \chi_T(v) = \chi_{\bar{T}}(v)$, so $e \in \chi_{\bar{T}}(v)$. Since this is true for any hyperedge $e$, the condition holds.
2. Since we constructed $\bar{T}$ by removing a node from $T$ and connecting its neighbors, it follows that if the path from node $x$ to node $y$ in $\bar{T}$ passes through a node $z$, then the path from $x$ to $y$ in $T$ also passes through $z$. The condition then follows directly from condition (2) of $T$.
3. The argument here is the same as the argument for condition (1).
4. This condition follows directly from transitivity of the subset relation.

Therefore, $\bar{T}$ is a hierarchy decomposition of $G$. To show that $\mathsf{w}(T) = \mathsf{w}(\bar{T})$, it suffices to notice that removing node $u$ from $T$ can't change its width, since node $v$ has at least as many hyperedges in its labeling as $u$. This proves the lemma. □

**Lemma 13.** *Let $T$ be a hierarchy decomposition of a hypergraph $G$. Let $u$ be a node of $T$, and let $v$ be a child of $T$ such that $\chi_T(u) = \chi_T(v)$. Let $\bar{T}$ be the graph minor of $T$ that is formed by identifying $u$ and $v$ by contracting along their connecting edge. Assume that the nodes $\bar{T}$ have the same labeling as the corresponding nodes in $T$, and that the new node, $w$, has $\chi_{\bar{T}}(w) = \chi_T(u) = \chi_T(v)$. Then $\bar{T}$ is a hierarchy decomposition of $G$, and*

$$\mathsf{w}(T) = \mathsf{w}(\bar{T}).$$

*Proof.* We validate the conditions of the definition of hierarchy decomposition individually.

1. The set of labelings of $\bar{T}$ is exactly the same as the set of labelings of $T$. Since $T$ is a hierarchy decomposition of $G$, the condition follows directly from condition (1) of $T$.
2. Similarly, the set of labelings of any path in $\bar{T}$ is the same as the set of labeling of the corresponding path in $T$. The condition then follows directly from condition (2) of $T$.
3. The argument here is the same as the argument for condition (1).
4. The labeling of every node's parent in $\bar{T}$ will be the same as the labeling of that node's parent in $T$. Therefore, the condition follows from condition (3) of $T$.

Therefore, $\bar{T}$ is a hierarchy decomposition of $G$. To show that $\mathsf{w}(T) = \mathsf{w}(\bar{T})$, it suffices to notice the set of labelings of nodes of $\bar{T}$ is the same as the set of labelings of nodes of $T$; therefore they must have the same width. $\qquad\square$

**Lemma 14.** *For any factor graph $G$ and for any hierarchy decomposition $T$ of $G$,*
$$\mathsf{w}(T) \geq \mathsf{hw}(G).$$

*Proof.* As in the proof of Lemma 9, we will prove this result by multiple induction. In what follows, we assume that the statement holds for all graphs with either fewer vertexes and no more hyperedges than $G$, or fewer hyperedges and no more vertexes than $G$. We also assume that for a particular $G$, the statement holds for all hierarchy decompositions of $G$ that have fewer nodes than $T$.

There are five possibilities:

1. The root $r$ of $T$ is labeled with $\chi(r) = \emptyset$, and $r$ has no children.
2. The root $r$ of $T$ is labeled with $\chi(r) = \emptyset$, and $r$ has one child.
3. The root $r$ of $T$ is labeled with $\chi(r) = \emptyset$, and $r$ has two or more children, at least one of which, $x$, is also labeled with $\chi(x) = \emptyset$.
4. The root $r$ of $T$ is labeled with $\chi(r) = \emptyset$, and $r$ has two or more children, none of which are labeled with the empty set.
5. The root $r$ of $T$ is labeled with $\chi(r) \neq \emptyset$.

We consider these cases separately.

**Case 1**  If $r$ is labeled with the empty set and has no children, then it follows from condition (1) that there are no hyperedges in $G$. Therefore, $\mathsf{hw}(G) = 0$. Also, since $r$ is labeled with the empty set, $\mathsf{w}(T) = 0$. So, the statement holds in this case.

**Case 2**  If $r$ has exactly one child, then by Lemma 12, there is a hierarchy decomposition $\bar{T}$ of $G$ such that $\bar{T}$ has fewer nodes than $T$ and $\mathsf{w}(T) = \mathsf{w}(\bar{T})$. By the inductive hypothesis,
$$\mathsf{w}(\bar{T}) \geq \mathsf{hw}(G).$$

So, the statement holds in this case.

**Case 3**  If $r$ is labeled with the empty set and has two or more children, at least one of which is also labeled with the empty set, then by Lemma 13, there is a hierarchy decomposition $\bar{T}$ of $G$ such that $\bar{T}$ has fewer nodes than $T$ and $\mathsf{w}(T) = \mathsf{w}(\bar{T})$. By the inductive hypothesis,
$$\mathsf{w}(\bar{T}) \geq \mathsf{hw}(G).$$

So, the statement holds in this case.

**Case 4**  Consider the case where $r$ is labeled with the empty set and has two or more children, none of which are labeled with the empty set. Let $T_1, T_2, \ldots, T_l$ be the labeled subtrees rooted at the children of $r$. Notice that if a hyperedge $e$ appears in the labeling of some node of $T_i$, then it can't appear in the labeling of any node of $T_j$ for $i \neq j$, since otherwise by condition (2) it must appear in the labeling of the root node $r$, and $r$ has an empty labeling. Therefore, the hyperedges of $G$ are partitioned among the subtrees $T_i$. Furthermore, condition (3) implies that hyperedges associated with different subtrees are disconnected, and therefore that graph $G$ is disconnected. Therefore, if we let $G_i$ denote the $m$ connected components of $G$.
$$\mathsf{hw}(G) = \max_{i \leq m} \mathsf{hw}(G_i).$$

Now, let $\bar{G}_i$, for $i \leq l$, denote the subgraph of $G$ that consists of the nodes that are connected to a hyperedge in a labeling of a node of $T_i$. Clearly, the $\bar{G}_i$ will be disconnected, so

$$\mathsf{hw}(G) = \max_{i \leq l} \mathsf{hw}(\bar{G}_i).$$

Each $\bar{G}_i$ will also have fewer vertexes and hyperedges than $G$, so by the inductive hypothesis,

$$\mathsf{w}(T_i) \geq \mathsf{hw}(\bar{G}_i).$$

Therefore,

$$\mathsf{w}(T) = \max_{i \leq l} \mathsf{w}(T_i) \geq \max_{i \leq l} \mathsf{hw}(\bar{G}_i) = \mathsf{hw}(G);$$

so the statement holds in this case.

**Case 5**  Finally, consider the case where the root node $r$ is labeled with some hyperedge $e$. Let $\bar{G}$ be the graph that results from removing $e$ from $G$, and let $\bar{T}$ be the hierarchy decomposition that results from removing $e$ from all the labellings of $T$. Clearly, $\bar{T}$ will be a hierarchy decomposition of $\bar{G}$. Furthermore, $\bar{G}$ has fewer hyperedges and the same number of vertexes as $G$, so by the inductive hypothesis,

$$\mathsf{w}(\bar{T}) \geq \mathsf{hw}(\bar{G}).$$

Furthermore, since removing $e$ decreases the size of each of the labelings of $\bar{T}$ by 1,

$$\mathsf{w}(\bar{T}) = \mathsf{w}(T) - 1.$$

Therefore,

$$\mathsf{w}(\bar{T}) \geq \mathsf{hw}(\bar{G}) + 1 \geq \mathsf{hw}(G),$$

so the statement holds in this case.

Therefore, the statement holds in all cases, so the lemma follows from induction on all hypergraphs and all hierarchy decompositions. □

**Lemma 15.** *For any factor graph $G$, there exists a hierarchy decomposition $T$ of $G$ such that*

$$\mathsf{w}(T) = \mathsf{hw}(G).$$

*Proof.*  As in the proof of Lemma 9, we will prove this result by multiple induction. In what follows, we assume that the statement holds for all graphs with either fewer vertexes and no more hyperedges than $G$, or fewer hyperedges and no more vertexes than $G$.

There are three possibilities:

1. $G$ has no hyperedges.
2. $G$ is disconnected.
3. $G$ is connected.

We consider these cases separately.

**Case 1**  If $G$ has no hyperedges, then the tree $T$ with a single node labeled with the empty set is a hierarchy decomposition for $G$. It will satisfy

$$\mathsf{w}(T) = 0.$$

Also, by (4), for the case with no hyperedges,

$$\mathsf{hw}(G) = 0,$$

so the statement holds in this case.

**Case 2**  Assume that $G$ is disconnected, and its connected components are $G_i$. Then by (3),

$$\mathsf{hw}(G) = \max_i \mathsf{hw}(G_i).$$

Since each $G_i$ has fewer nodes and hyperedges than $G$, by the inductive hypothesis there exists a hierarchy decomposition $T_i$ for each $G_i$ such that

$$\mathsf{w}(T_i) = \mathsf{hw}(G_i).$$

Let $T$ be the tree that has a root node labeled with the empty set, and where the subtrees rooted at its children are exactly the $T_i$ above. We now show that $T$ is a hierarchy decomposition for $G$ by validating the conditions.

1. Any hyperedge $e$ of $G$ must be a hyperedge of exactly one $G_i$. Since $T_i$ is a hierarchy decomposition of $G_i$, by condition (1) it follows that $e$ appears in the labeling of some node of $G_i$. Therefore, $e$ will appear in the labeling of the corresponding node of $G$, and the condition holds.
2. Again, any hyperedge $e$ of $G$ must be a hyperedge of exactly one $G_i$. Therefore, if $e$ appears in two nodes of $T$, those two nodes must both be part of the same subtree $T_i$. The condition follows from the corresponding condition of $T_i$.
3. Since $G$ is disconnected, any pair of hyperedges in $G$ that share a vertex must both be part of exactly one $G_i$. The condition then follows from the corresponding condition of $T_i$.
4. This condition follows directly from the fact that for any $X$, $\emptyset \subseteq X$, and from the corresponding condition for the subgraphs $T_i$.

Therefore $T$ is a hierarchy decomposition for $G$. Furthermore, since the labelings of nodes of $T$ are the union of the labelings of nodes of $T_i$,

$$\mathsf{w}(T) = \max_i \mathsf{w}(T_i).$$

Therefore,

$$\mathsf{hw}(G) = \max_i \mathsf{hw}(G_i) = \max_i \mathsf{w}(T_i) = \mathsf{w}(T),$$

so the statement holds in this case.

**Case 3**  Assume that $G$ is connected. Then by (2),

$$\mathsf{hw}(G) = 1 + \min_{e \in E} \mathsf{hw}(\langle N, E - \{e\}\rangle).$$

Let $b$ be a hyperedge that minimizes this quantity, and let $\bar{G}$ be the graph that results from removing this hyperedge from $G$. Then,

$$\mathsf{hw}(G) = 1 + \mathsf{hw}(\bar{G}).$$

Now, $\bar{G}$ has fewer hyperedges than $G$, so by the inductive hypothesis, there exists a hierarchy decomposition $\bar{T}$ of $\bar{G}$ such that

$$\mathsf{hw}(\bar{G}) = \mathsf{w}(\bar{T}).$$

Let $T$ be the tree that results from adding edge $b$ to every labeling of a node of $\bar{T}$. Clearly,

$$\mathsf{w}(T) = \mathsf{w}(\bar{T}) + 1.$$

We now show that $T$ is a hierarchy decomposition for $G$ by validating the conditions.

1. Any hyperedge $e$ of $G$ is either hyperedge $b$ or some hyperedge in $\bar{G}$. If it is $b$, then it must appear in a labeling of a node of $T$ since it appears in all such labelings. Otherwise, the condition follows from the corresponding condition of $\bar{T}$.
2. Again, any hyperedge $e$ of $G$ is either hyperedge $b$ or some hyperedge in $\bar{G}$. If it is $b$, then the condition follows from the fact that all node labelings of $T$ contain $b$. Otherwise, the condition follows from the corresponding condition of $\bar{T}$.
3. For any pair of edges $(e, f)$ of $G$, either $b \notin \{e, f\}$, or $b \in \{e, f\}$. If $b \in \{e, f\}$, then the condition follows from condition (1) of $\bar{T}$ and the fact that all node labelings of $T$ contain $b$; otherwise, the condition follows from the corresponding condition of $\bar{T}$.
4. This condition follows directly from the fact that for any $X$ and $Y$, if $X \subseteq Y$, then $X \cup \{b\} \subseteq Y \cup \{b\}$.

Therefore $T$ is a hierarchy decomposition for $G$, and

$$\mathsf{w}(T) = \mathsf{w}(\bar{T}) + 1 = \mathsf{hw}(\bar{G}) + 1 = \mathsf{hw}(G),$$

so the statement holds in this case.

Therefore the statement holds in all cases, so the lemma follows from induction over all factor graphs. ☐

Next, we restate and prove Statement 9.

**Statement 9.** *The* hierarchy width *of a hypergraph $G$ is equal to the minimum width of a hierarchy decomposition of $G$.*

*Proof.* This statement follows directly from Lemmas 14 and 15. ☐

Now, we provide a definition for hypertree width, and prove the comparison statements from the body of the paper.

**Definition 12** (Hypertree Decomposition [6])**.** A *hypertree decomposition* of a hypergraph $G$ is a structure $\langle T, \rho, \chi \rangle$, where $\langle T, \rho \rangle$ is a tree decomposition for the primal graph of $G$, $\chi$ is a labeling of the nodes of $T$ with hyperedges of $\chi$, and the following conditions are satisfied: (1) for any node $u$ of $T$, and for all vertices $x \in \rho(u)$, there exists an edge $e \in \chi(u)$ such that $x \in e$; and (2) for any node $u$ of $T$, for any edge $e \in \chi(u)$, and for any descendant $v$ of $u$ in $T$, $e \cap \rho(v) \subseteq \rho(u)$.

The *width* of a hypertree decomposition, denoted $\mathsf{w}(T)$, is defined (as in the hierarchy decomposition case) to be the cardinality of the largest hyperedge-labeling of a node of $T$. That is,

$$\mathsf{w}(T) = \max_u |\chi(u)|.$$

The *hypertree width* $\mathsf{tw}(G)$ of a graph $G$ is the minimum width among all hypertree decompositions of $G$.

We restate and prove Statement 1.

**Statement 1.** *For any factor graph $G$, $\mathsf{tw}(G) \leq \mathsf{hw}(G)$.*

*Proof.* Consider the hierarchy decomposition $T$ of $G$ that satisfies $\mathsf{w}(T) = \mathsf{hw}(G)$. Assume that we augment $T$ with an additional node labeling $\rho(v)$ that consists of the vertexes of $G$ that are part of at least one edge of $\chi(v)$ (that is, $\rho(v) = \cup \chi(v)$). We show that $T$ is a hypertree decomposition of $G$.

First, we must show that $\langle T, \rho \rangle$ is a tree decomposition for the primal graph of $G$; we do so by verifying the conditions independently.

1. Each vertex of $G$ must appear in the labeling of some node of $T$, since (by assumption) it must appear in some hyperedge of $G$, and each hyperedge appears in the labeling of some node of $T$ because $T$ is a hierarchy decomposition of $G$.
2. Similarly, any two vertexes $(x, y)$ that are connected by an edge in the primal graph of $G$ must appear together in some hyperedge of $G$. Since each hyperedge of $G$ appears in the edge-labeling of some node of $T$, it follows that $x$ and $y$ must both appear in the vertex-labeling of that node.
3. Finally, assume that vertex $x$ appears in the labeling of two different nodes $u$ and $v$ of $T$. Since $x \in \rho(u)$, there must exist an edge $e \in \chi(u)$ such that $x \in e$; similarly, there must be an edge $f \in \chi(v)$ such that $x \in f$. By condition (3) of the definition of hierarchy decomposition, there must exist a node $w$ of $T$ such that $\{e, f\} \subseteq \chi(w)$. By condition (2) of the definition of hierarchy decomposition, for all nodes $z$ on the path between $u$ and $w$, $e \in \chi(z)$; similarly, for all nodes $z$ on the path between $v$ and $w$, $f \in \chi(z)$. Now, any node $z$ on the path between $u$ and $v$ must be either part of the path between $u$ and $w$ and the path between $v$ and $w$, therefore either $e \in \chi(z)$ or $f \in \chi(z)$. It follows that $x \in \rho(z)$, as desired.

Therefore $T$ is a tree decomposition for the primal graph of $G$.

Next, we show that the $T$ is also a hypertree decomposition for $G$ by verifying the conditions independently.

1. For any node $u$ of $T$, and for all vertices $x \in \rho(u)$, there must exist an edge $e \in \chi(u)$ such that $x \in e$, because $\rho(u) = \cup \chi(v)$. Therefore the condition holds.
2. For any node $u$ of $T$ and for any descendant $v$ of $u$ in $T$, we know from condition (4) of the definition of hierarchy decomposition that $\chi(u) \subseteq \chi(v)$. So, for any edge $e \in \chi(u)$, it also holds that $e \in \chi(v)$. Therefore, $e \cap \rho(v) = e \subseteq \rho(u)$, and so the condition holds.

We conclude that $T$ is a hypertree decomposition for $G$. But, its width is $\mathsf{hw}(G)$. Since the hypertree width of $G$ is the minimum width among all hypertree decompositions of $G$, it follows that

$$\mathsf{tw}(G) \leq \mathsf{hw}(G),$$

which is the desired expression. $\square$

Next, we restate and prove Statement 2.

**Statement 2.** *For any fixed $k$, computing whether $\mathsf{hw}(G) \leq k$ can be done in time polynomial in the number of factors of $G$.*

*Proof.* Consider Algorithm 1, which computes this quantity for a fixed $k$.

---

**Algorithm 1** $\text{HW}(G, k)$: Compute if $\text{hw}(G) \leq k$

---

  **if** $G$ has no edges **then**
    **return** true
  **end if**
  **if** $k = 0$ **then**
    **return** false
  **end if**
  **for** $e \in \text{edges}(G)$ **do**
    Let $\bar{G}$ be the graph that results from removing $e$ from the connected component of $G$ that contains $e$
    **if** $\text{HW}(\bar{G}, k - 1)$ **then**
      **return** true
    **end if**
  **end for**
  **return** false

---

Each execution of $\text{HW}$ requires at most linear time in the number of hyperedges of $G$ to compute its connected components. If we let $e$ be the number of hyperedges of $G$, it also requires at most $e$ executions of $G$ with parameter $k - 1$. Therefore, $\text{HW}$ will run in time $O(e^k)$, which is polynomial in the number of hyperedges, as desired. $\qquad\square$

As an aside here, we note that the hierarchy width can be expressed in terms of an existing graph parameter, the *tree-depth* [29, p. 115], which we denote $\text{td}(G)$. To do this, we let $L(G)$, the *line graph* of $G$, denote the graph such that every factor of $G$ is a node of $L(G)$, and two nodes of $L(G)$ are connected by an edge if and only if their corresponding factors share a dependent variable. Using this, it is trivial to show (by definition) that

$$\text{hw}(G) = \text{td}(L(G)).$$

Since it is a known result that it is possible to compute whether a graph has tree-depth at most $k$ in time polynomial (in fact, linear) in the number of nodes of the graph, we could have also used this to prove Statement 2; we avoided doing this to make the result more accessible.

**Statement 10.** *The hierarchy width of a factor graph $G$ is greater than or equal to the maximum degree of a variable in $G$.*

Finally, we restate and prove Statement 10.

**Statement 10.** *The hierarchy width of a factor graph $G$ is greater than or equal to the maximum degree of a variable in $G$.*

*Proof.* The statement follows by induction. Considering the parts of the definition of hierarchy width individually, removing a single hyperedge from a hypergraph only decreases the maximum degree of the hypergraph by at most 1, and splitting the hypergraph into connected components doesn't change the maximum degree. $\qquad\square$

# D  Proofs of Factor Graph Template Results

In this section, we prove the results in Section 3.1. First, we prove Lemma 16.

**Lemma 16.** *If $G$ is an instance of a hierarchical factor graph template $\mathcal{G}$ with $E$ template factors, then $\text{hw}(G) \leq E$.*

**Lemma 16.** *If $G$ is an instance of a hierarchical factor graph template $\mathcal{G}$ with $E$ template factors, then $\text{hw}(G) \leq E$.*

*Proof.* We prove this result using the notion of hierarchy decomposition from the previous section. For the instantiated factor graph $G$, let $T$ be a tree where each node $v$ of $T$ is associated with a particular assignment of the first $n$ head symbols of the rules. That is, there is a node of $T$ for

each tuple of object symbols $(x_1, \ldots, x_m)$—even for $m = 0$. (Since the rules are hierarchical, and therefore must contain the same symbols in the same order, these assignments are well-defined.)

Label each node $v$ of $T$ with the set containing all the instantiated factors of $G$ that agree with this assignment of head variables. That is, if the head variable assignment for a factor $\phi$ is $(y_1, \ldots, y_p)$, then it will be in the labeling of the node $v = (x_1, \ldots, x_m)$ if and only if $m \geq p$ and for all $i \leq p$, $y_i = x_i$. It is not hard to show that this is a valid hierarchy decomposition for this factor graph.

The width of this hierarchy decomposition is at most the number of template factors $E$ because each node $v$ can only possibly contain a single instantiated factor for each template factor. (This is because for each node, only one possible assignment of head variables is compatible with that node.) The Lemma now follows from an application of Statement 9.  $\square$

Next, we restate and prove Statement 4.

**Statement 4.** *For any fixed hierarchical factor graph template $\mathcal{G}$, if $G$ is an instance of $\mathcal{G}$ with bounded weights using either logical or ratio semantics, then the mixing time of Gibbs sampling on $G$ is polynomial in the number of objects $n$ in its dataset. That is, $t_{\mathrm{mix}} = O\left(n^{O(1)}\right)$.*

*Proof.* If we use either logical or ratio semantics, the maximum factor weight will be bounded with $M = O(\log n)$. We furthermore know from Lemma 16 that for a particular hierarchical template, the hierarchy width is bounded independent of the dataset. So, $hM = O(\log n)$, and the Statement now follows directly from an application of Theorem 2  $\square$

## Secondary Literature

[29] Patrice Ossona de Mendez et al. *Sparsity: graphs, structures, and algorithms*. Springer Science & Business Media, 2012.

[30] Frank den Hollander. *Probability Theory: The Coupling Method*. 2012.