[Reviews · NeurIPS 2015]

Submitted by Assigned_Reviewer_1

This paper develops a new property of hypergraphs that helps bound the mixing time in Gibbs sampling. Analyzing the mixing time is always a hard problem in MCMC, so this paper addresses an important issue in Gibbs sampling. The paper defines the notion of hierarchy width for factor graphs. The main result that the authors show is that, the mixing time is exponential in the hierarchy-width of the factor graph and the maximum difference in weights encoded by the potentials of the factor graph. The authors then show that for specific factor graph templates, the aforementioned results can be applied and the mixing time of Gibbs sampling can be bounded polynomially in the number of variables that instantiate the templates.

The main novelty is theoretical results that show that for specific classes of template factor graphs, Gibbs sampling mixes rapidly.

Figure 1 seems to come out of the blue. It is hard to understand this figure since almost everything in this figure is explained much later.

One of the main problems with Gibbs sampling is mixing when there are 0 values (or near-determinism) in the potentials. I was not sure how your results would change in such cases. Are there some assumptions about the underlying distribution structure.

What is the treewidth typically when factor graph templates are hierarchical? That is, can we have high treewidth models using these templates? If not, then it might not require approximate inference such as Gibbs sampling.

I was a little confused by what the three semantics represent. That is, is there a drawback from using the logical/ratio semantics (where your results hold) over the linear semantics (where your results do not hold)

On the real world applications, if only subgraphs are hierarchical do your results of rapid mixing hold? I assumed that you need the entire factor graph template to be hierarchical as in section 3.1. Does this happen often in practice?
Summary: This paper proves that on certain template factor graphs with hierarchical structure Gibbs sampling provably mixes in polynomial time. I think the paper is mostly well-written and handles a tricky problem. The experiments section could be expanded with some details regarding applicability to real-world applications

Submitted by Assigned_Reviewer_2

This paper proposed "hierarchy width" which is a new graph property for bounding the mixing time of Gibbs sampling on factor graphs.

The reviewer concerns about the fact that the hierarchy width of a graph is always larger than its hypertree width.

By my intuition, polynomial time exact inference seems more difficult that polynomial mixing time sampling.

But the fact is counterintuitive.

Why sampling is much harder than exact inference?
Summary: It is one of my light review paper. This paper proposed a new graph property "hierarchy width", which bound the mixing time of Gibbs sampling on factor models.

The theoretical result of this paper seems important for understanding the property of Gibbs sampling on factor graphs.

Submitted by Assigned_Reviewer_3

This paper presents a new upper bound for the mixing time of Gibbs sampling.

Unlike previous specialized results, the bound applies to arbitrary discrete factor graphs.

The bound is too loose to be used in a quantitative way, but the paper uses the bound qualitatively to explain why certain real-world networks mix well.

The paper is well-written and covers a nice assortment of methods for proving mixing time (though most of the good stuff is in the appendix).

Section 4 shows that the mixing time of Ising models is related to hierarchy width.

This is again a qualitative, not quantitative, assessment of the theory.

The theorem from [10], p201 suggests that the maximum degree is the relevant quantity, rather than hierarchy width.

How does the figure look with maximum degree on the axis?

The paper claims that exact inference is not feasible in the real-world networks (as mentioned on line 418), which suggests that they have high tree-width.

The paper should report the tree-width and compare this to hierarchy-width and factor weight.

Does Theorem 2 give meaningful bounds for the real-world networks?

The related work section is a bit odd since most of the papers cited are not about mixing time.

It would make more sense to cite previous work on bounds for mixing time in factor graphs.

In particular, there are existing bounds for special graphs that are much tighter than Theorem 2.

One such bound was used by Liu and Domke (2014).

"Projecting Markov Random Fields for Fast Mixing" Xianghang Liu, Justin Domke NIPS 2014 http://arxiv.org/abs/1411.1119

Section 2.1 has 'q' instead of 'Q' in the equations, and typo 'figure 2(a)' instead of 'figure 2(b)'.
Summary: New theory about Gibbs sampling, presented in a readable and thought-provoking way, though only used in a qualitative sense.

Submitted by Assigned_Reviewer_4

The paper makes two main contributions: (1) A novel notion of hierarchy width that can be used to bound the mixing time of Gibbs sampling in factor graphs (along with the requirement that the parameters of the factor graph are bounded). (2) A new relational modeling language for generating factor graphs on which Gibbs sampling will mix rapidly.

Another minor contribution is empirical evaluation, which shows that the theoretical results are indeed applicable to real-world problems.

Overall, the paper is well-written and was a joy to read. I don't have any special comments except that it might be instructive to also look at "Blocked Gibbs sampling" as well as "Rao-Blackwellised Gibbs sampling" and see how these can help reduce the hierarchy width. On the flip side, can hierarchy width be used as a strategy for constructing blocks and for Rao-Blackwellising the Gibbs sampler. See for example, Venugopal and Gogate, UAI 2013; Hamze and de Freitas, UAI 2004; and Bidyuk and Dechter, Journal of AI Research, 2007.
Summary: The paper introduces a novel notion of hierarchy width of (relational) factor graphs and shows that along with a bounded parameters requirement it can be used to derive novel bounds on the mixing time of Gibbs sampling. It is well-written, makes an important advance and has good experimental results.

Author Feedback
Author rebuttal: We thank all the reviewers for their time in giving us reviews and feedback.

One point deserves specific comment: R1, R2, and R4 all had questions about the relationship between hypertree width and hierarchy width, and how this relates to the comparison between Gibbs sampling and exact inference techniques.

When hierarchy width is bounded, the hypertree width is similarly bounded (Statement 1 in our paper). This means that for the models we focus on, where Gibbs mixes in polynomial time, exact inference also runs in polynomial time. However, for graphs with sufficiently small weights (such as the Paleontology model we mention), the polynomial exponent for Gibbs will be smaller than for exact sampling. This means that exact sampling is not feasible for some graphs for which Gibbs sampling is feasible. In addition to this, Gibbs sampling is a practical and easy-to-apply algorithm used heavily in a wide variety of applications, so it interesting to study its performance, regardless of how exact inference might perform on the same graph. We agree that our presentation of this comparison needs improvement, and we will add prose to our paper to make it more clear.

Reviewer 1

R1 notes that "Figure 1 seems to come out of the blue." To reduce confusion, we will move this figure to later in the paper, before the experiments section.

When there are 0 values, representing near-determinism, in the potentials, these sorts of Gibbs sampling mixing results break down. There are very simple cases of distributions with 0-probability events for which Gibbs sampling does not converge at all. This is a well-known problem: a wide variety of approximate inference models (loopy BP, simulated tempering, etc) fail to handle deterministic dependencies. However, we can handle the case where the values of some of the variables are known deterministically: in this case our results still hold.

There is not necessarily a drawback from using logical/ratio semantics; it simply samples from a different distribution. Depending on the application, logical/ratio semantics may be better or worse than linear. In one of the papers we cite, "Incremental knowledge base construction using deepdive," the authors exhibit some practical cases in which logical/ratio semantics produce superior results; in all five of the KBC workloads they study, using ratio semantics resulted in an F1 score that was no worse than the other semantics (their Figure 10b).

We do need the entire factor graph template to be hierarchical in order to apply Lemma 7 (although, a non-hierarchical template can still produce graphs with bounded hierarchy width). This happens with some frequency in practice, although it by no means covers all factor graphs that are of interest.

Reviewer 2

R2 notes that, for Figure 5(a), "the maximum degree is the relevant quantity, rather than hierarchy width." Figure 5(a) would look similar were we to use maximum degree on the x axis rather than hierarchy width, since for the Ising models we used, maximum degree and hierarchy width are related quantities.

R2 asks about the feasibility of exact inference in real-world networks. All the models we analyze which have bounded hierarchy width also have bounded treewidth; this is generally true (Statement 1). However, it may still be possible that exact inference is not feasible to run in practice, even for graphs of bounded treewidth, because of the large (but still polynomial-time) runtime involved.

We will augment our related work section by citing "Projecting Markov Random Fields for Fast Mixing," as suggested by R2, in addition to other relevant papers about mixing times.

We will fix the errors R2 outlines in Section 2.1.

Reviewer 3

We thank R3 for their perceptive review of our paper. We also thank R3 for suggesting that we look at "Blocked Gibbs sampling" and "Rao-Blackwellised Gibbs sampling" and giving examples of useful papers in these areas. Although they seem to be out of the scope of this paper, we are excited to investigate these modified Gibbs sampling algorithms as future work for applying hierarchy width in this area.

Reviewer 4

R4 asks why the hierarchy width is larger than the hypertree width, which seems to imply that sampling is much harder than exact inference. To gain intuition about this, note that there are some classes of graphs, for example the voting model with linear semantics, on which Gibbs sampling mixes in exponential time, but exact inference methods can solve in polynomial time. There are other classes of graphs, for example unbounded treewidth graphs with very small factor weights, for which exact inference will take exponential time to run but Gibbs sampling will mix in polynomial time. So, really, neither one of sampling nor exact inference is strictly "harder" than the other.

Review 5

We thank R5 for their review.

Review 6

We thank R6 for their review.